# Hamiltonian Variational Formulation of Three-Dimensional, Rotational Free-Surface Flows, with a Moving Seabed, in the Eulerian Description

**Constantinos P. Mavroeidis** 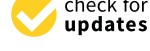 and **Gerassimos A. Athanassoulis** *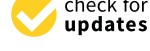

School of Naval Architecture and Marine Engineering, National Technical University of Athens, 15773 Athens, Greece
* Correspondence: makathan@gmail.com

**Abstract:** Hamiltonian variational principles have provided, since the 1960s, the means of developing very successful wave theories for nonlinear free-surface flows, under the assumption of irrotationality. This success, in conjunction with the recognition that almost all flows in the sea are not irrotational, raises the question of extending Hamilton's principle to rotational free-surface flows. The Euler equations governing the bulk fluid motion have been derived by means of Hamilton's principle since the late 1950s. Nevertheless, a complete variational formulation of the rotational water-wave problem, including the derivation of the free-surface boundary conditions, seems to be lacking until now. The purpose of the present work is to construct such a missing variational formulation. The appropriate functional is the usual Hamilton's action, constrained by the conservation of mass and the conservation of fluid parcels' identity. The differential equations governing the bulk fluid motion are derived as usually, applying standard methods of the calculus of variations. However, the standard methodology does not provide enough structure to obtain the free-surface boundary conditions. To overcome this difficulty, differential-variational forms of the aforementioned constraints are introduced and applied to the boundary variations of the Eulerian fields. Under this transformation, both kinematic and dynamic free-surface conditions are naturally derived, ensuring the Hamiltonian variational formulation of the complete problem. An interesting feature, appearing in the present variational derivation, is a dual possibility concerning the tangential velocity on the boundary; it may be either the same as in irrotational flow (no condition) or zero, corresponding to the small-viscosity limit. The deeper meaning and the significance of these findings seem to deserve further analysis.

**Keywords:** Hamilton's principle; free-surface flows; rotational flows; boundary conditions; variational formulation; Clebsch potentials

## 1. Introduction

### 1.1. Motivation

The significance of understanding and predicting phenomena related to the wave motion in the sea can hardly be overestimated nowadays. It may be sufficient to mention that it is of focal interest in naval, coastal, and offshore engineering, and also of great importance for oceanography. A distinctive feature of these nonlinear waves is the presence of the unknown free surface, which has to be predicted simultaneously with the underlying wave field. This feature greatly complicates both the mathematical analysis and the numerical computation of free-surface problems.

It turns out that these difficulties become milder and easier to handle under specific mathematical formulations. For example, in the case of an ideal (inviscid) incompressible liquid undergoing an irrotational wave motion, the Hamiltonian variational formulation, initiated by Petrov (1964) [1] and Zakharov (1968) [2], has produced effective equations for studying non-linear waves and very efficient schemes for numerical computations. The

closely related unconstrained variational principle of Luke (1967) [3] facilitated further development of nonlinear wave theories and numerical techniques. In the last 60 years, a significant amount of works have developed and exploited the Hamiltonian formulation of the irrotational water-wave problem, and its ramifications (Some indicative recent works on this topic are [4–9], and the references therein. Readers interested in additional relevant material may also resort to an internet search, using key phrases such as "Hamiltonian approach to water waves").

However successful the irrotational model may be, it is well known that rotationality is always present in the sea. Additionally, this is true not only for phenomena such as wave breaking and air–sea interaction but also for mild realistic sea waves. Oceanographers introduced the concept of the wave-induced non-breaking turbulence in 2004 [10,11] and found significant improvements in upper-ocean circulation predictions when they incorporated it in general circulation models [12]. The authors of the latter paper, trying to explain the scarcity of rotational wave theories, argue that "*the success of potential theories of nonlinear waves . . . made the applications of non-potential wave theories seem redundant and eventually even led to them being nearly forgotten*".

### 1.2. History and Background Literature

The need for rotational wave theories, in combination with the extraordinary success of the Hamiltonian variational formulation for irrotational waves, raises the question of extending the method to the more realistic (and more complicated) case of rotational free-surface flows. In this case, however, the choice between Lagrangian and Eulerian description of the fluid motion becomes significant. In the second approach, the physical fields, that is, the velocity $u$, density $\rho$, and pressure $p$, are expressed as functions of spatial coordinates $(x, z) = (x_1, x_2, z)$ and time $t$ while in the former one, the main field is the position of fluid parcels, $X(a, t)$, considered as a function of their initial positions $a = (a_1, a_2, a_3)$ and the time. Thus, it is clear that the Lagrangian description, being inconvenient in applications and almost forgotten in engineering hydrodynamics, is directly compatible with the methods of analytical mechanics, based on virtual displacements. Lagrange himself derived the hydrodynamic equations (in the Lagrangian description) by using the D'Alembert principle of virtual work in 1815 (Mécanique Analytique, Tome 2, Section 11). Nevertheless, the derivation of Euler's equations from Hamilton's principle had to wait for more than a century before appearing in 1929 [13]. In the present paper, we are primarily interested in the variational formulation of rotational flows in their Eulerian description; the Lagrangian one will be touched upon only at the extent that it is necessary for developing the former.

The direct application of Hamilton's principle to the derivation of equations governing rotational flows in the Eulerian description stumbles upon a fundamental controversy. The validity of Hamilton's principle is crucially dependent on applying virtual variations of the positions of fluid parcels, $X(a, t)$ for fixed $(a, t)$. However, fluid parcels' positions are completely absent in the Eulerian formulation, and the physical fields, e.g., $u = u(x, z, t)$, are defined in terms of the spatial variables. Accordingly, the natural variations of the involved fields, e.g., $\delta u(x, z, t)$, are variations for fixed $(x, z, t)$. The situation described above will be referred to, followingly, as the *variational controversy* (of Eulerian fluid dynamics).

The first attempt to derive the Eulerian equations of fluid dynamics by means of Hamilton's principle was carried out by Herivel in 1955 [14]. He recognized the need for the introduction of constraints in the standard Lagrangian function (kinetic minus potential energy), and as such he implemented the conservation of mass and the conservation of entropy. The obtained variational equations apparently produce Euler's equation of momentum; however, the underlying velocity representation turns out to be restricted in irrotational flows when the entropy is assumed to be constant, as it should (the variational controversy was not well understood in 1955); see also the discussion in [15] (p. 5). Let it be noted that, from a purely mechanical point of view, entropy should not be of significance for the dynamics of an ideal mechanical system, as the considered fluid flow is. The variational

controversy has been resolved, at least for the bulk motion of the fluid, by means of the clever proposal of Lin [16,17] around 1959, who introduced the purely mechanical constraint of the conservation of fluid parcels' identity (This proposal first appeared in [16], with reference to an unpublished note of Lin. A similar reference is given in [18]). See also Section 3.1. Although Lin's constraint is commonly used in conjunction with the entropy constraint [15,19], in fact, it can replace the latter, leaving us with a purely mechanical variational formulation.

The Herivel–Lin approach became the standard one after the publication of the works of Serrin [16] and Eckart [18]. Many papers have appeared since then, exploiting various aspects and deepening our understanding of this variational formulation. Penfield [20], Bretherton [21], and Salmon [22] discussed the importance of using Hamilton's principle (the Herivel–Lin approach) for studying Eulerian rotational flows of ideal fluids. The necessity of introducing Lin's constraint has also been highlighted by means of transformations between the two descriptions of fluid flow [22,23]. In an alternative direction, the Eulerian variational formulation was obtained from that in the Lagrangian description using canonical transformations [24,25]. The papers by Seliger and Whitham [15] and Fukagawa and Fujitani [19] dealt with many aspects of Hamilton's principle for rotational flows, including its relation with the Clebsch approach (see below, at the end of the Introduction). They also discussed the variational formulation with reduced versions of Lin's constraint, that is, velocity representation with fewer potentials; see, also, the discussion in [22] (Section 5). However, as first illustrated in [21], and later proved with more advanced mathematical tools in [26–28], these considerations apply only to a restricted class of flows, with zero helicity and without points of vanishing vorticity; see [29,30] (Section 6.17).

All papers mentioned above, dealing with the Herivel–Lin Hamiltonian variational formulation of three-dimensional (3D) rotational flows, *do not touch upon the issue of boundary conditions*. In fact, the only work that the present authors found with some discussion on boundary conditions in this context is the book by Berdichevsky [31] (Section 9.3), where he derives a form of the free-surface dynamic condition, having all the kinematic conditions a priori imposed. In a different direction, which uses Hamilton's principle in conjunction with the constrained variations of the Euler–Poincaré framework, a limited number of works dealing with free-surface boundary conditions have appeared recently (see [32,33], and references therein). Although these works are interesting and illuminating, they are limited to two-dimensional (2D) flows, utilizing a stream function, which drastically simplifies the treatment but is not applicable to 3D flows. Again, kinematic conditions need to be imposed as constraints in the variational formulation.

### 1.3. Contribution of the Present Paper

The purpose of the present paper is to provide a complete derivation, by means of Hamilton's principle, of the Eulerian equations of motion and the boundary conditions of 3D rotational water waves over a moving seabed. The novelty of the present paper lies in the consideration and the treatment of the boundary conditions. As a rule, works deriving the Euler equations variationally do not consider boundary conditions at all. In fact, the authors are not aware of any paper or book deriving both free-surface boundary conditions variationally in the present context. This means that rotational free-surface flows have not been given the status of a Hamiltonian system up to now. This is awkward, since this system is an ideal mechanical one, and, as such, it is expected to be Hamiltonian. It is also unfortunate since the Hamiltonian structure of a mechanical system permits the application of a manifold of methods and techniques for obtaining conservation laws, stability properties, simplified approximate model equations, deep theoretical results, and efficient numerical methods. The advantages of the Hamiltonian formulation are repeatedly mentioned by authors dealing with rotational flows without a free surface; see, e.g., [18,20,21,34,35] (A relevant - yet different - problem, where all these advantages have been fully exploited, is the irrotational water-wave problem, as mentioned at the beginning

of this Introduction). These advantages cannot be extended to the rotational water-wave problem until the latter is proved to be Hamiltonian as well.

**Remark 1.** It should be noted that the presence of a free surface is not a simple variation of the basic problem as, for example, is the presence of a moving rigid wall (with a predetermined velocity) instead of a non-moving rigid wall. The essential difference is that the free surface introduces a new unknown field, exhibiting its own dynamics, coupled with the dynamics of the substrate fluid. This deep difference between the two cases is well understood in irrotational flows. For example, in the case of incompressible irrotational flows, the equation governing non-free-surface problems is the Laplace equation while the equations governing the (nonlinear) water-wave problem (obtained using Hamilton's principle) are operator differential equations, in which the Laplace equation is used in an auxiliary way, just to define the coefficients of the equations governing the evolution of the free surface. See, for example, Equations (4a) and (4b) of [36].

To prove the Hamiltonian structure of the system under study, we follow the Herivel–Lin approach (the action functional is constructed using the standard Lagrangian, constrained by the conservation of mass and the conservation of fluid parcels' identity), and perform the variations using the standard methods of the calculus of variations. The volume-integral terms of the variational equation derive Euler's equation of motion, along with the Clebsch representation of the velocity field, as usual. Normally, from the boundary terms of the variational equation, it is expected to derive the boundary conditions. This seems not to be possible, however, if we consider the boundary variations of the involved fields to be independent of the boundary, as is the case within the fluid domain.

The fundamental contribution of this work is that it addresses this inconsistency from an entirely new point of view. Based on the argument (conjecture) that Lin's constraint, expressed by volume integrals, does not effectively act on the boundary, which is a lower-dimensional manifold, we use a stronger (locally valid) version of the corresponding constraints. The latter version is of a differential-variational nature, expressing the local variations of the Eulerian fields in terms of virtual displacements $\delta X$ of the fluid parcels. After substituting the variations of the Eulerian fields on the boundary with the corresponding expressions in terms of $\delta X$, the number of independent boundary terms decreases and their new forms, in conjunction with standard variational arguments, provide us with the expected boundary conditions on the free surface, on the moving seabed, and on any lateral rigid-wall boundary. These results can be considered as an a posteriori justification of our conjecture on the inadequacy of Lin's volume-integral constraint for the boundary conditions. An interesting and rather unexpected feature appearing in the present variational derivation is a dual possibility concerning the tangential velocity on the boundary. It may be either the same as in the irrotational flow (no condition) or zero, with the latter corresponding to the small viscosity limit. The deeper meaning and the significance of these findings seem to deserve further analysis [36].

The paper is organized as follows: In Section 2, we describe the geometry of the fluid domain, and present the usual differential formulation (governing equations and boundary conditions) of the problem. In Section 3, we present all the foundational aspects of the proposed variational formulation, which is based on Hamilton's action functional constrained by the conservation of mass and the conservation of parcels' identity (integral constraints). Then, we introduce the Lagrangian concept of virtual displacements, describing in detail why they are needed on the boundary, and produce the differential-variational constraints of the Eulerian variations, which are used subsequently. Having established the required quantities and concepts, we present the new variational principle in the form of a theorem, facilitating the understanding and the further use of the obtained result. We close Section 3 by presenting the steps of the proof of this variational formulation, which happens to be unconventional. The proof itself is the main objective of Sections 4 and 5. The general variational equation is derived in Section 4, and it is exploited to obtain the equations

of motion within the fluid domain, including explicit representations of the velocity and pressure fields in terms of Clebsch potentials. Section 5 is devoted to the treatment of the boundary variational equation. Its primitive form, as obtained by the standard approach of the calculus of variations, is reformulated using the differential-variational constraints. This new form permits us to derive the complete set of boundary conditions for each part of the boundary (free surface, moving seabed, fixed lateral rigid-wall boundaries). Last, in Section 6, we discuss the findings of this work and arrive at some concluding remarks.

*1.4. Related Research Directions That Are Not Considered in this Work*

In fact, the history of the variational formulation for unsteady rotational flows in the Eulerian description started in the 19th century, with the pioneering work of Clebsch in 1859 [37]. In this work, the author applies the theory of Pfaffian forms to obtain a representation of the velocity field in terms of three scalar functions (called Clebsch potentials) and observes that the hydrodynamic equations (expressed in terms of these potentials) may be obtained as the extremal condition of an integral functional. This functional is essentially the space-time integral of the pressure field. The main points of the Clebsch approach were summarized by Bateman (1929, 1944) [38,39] in a form adapted to compressible flows. Seliger and Whitham [15] and Fukagawa and Fujitani [18] have discussed the relation of the Clebsch approach with Hamilton's principle, without considering boundary conditions. Luke [3] provided a short note suggesting the possibility of exploiting the Clebsch–Bateman variational principle for barotropic free-surface flows, and this suggestion was recently elaborated further by Timokha [40]. In this approach, the dynamic free-surface condition is easily obtained because the Lagrangian density is just the pressure field. We do not follow this line of thought in the present paper, choosing Hamilton's principle instead, as a basis of our search for the variational formulation. This provides a direct connection with the foundational ground of analytical mechanics (canonical transformations, Noether's theorem, etc.).

Another direction that is not considered herein (with the exception of the two recent papers [32,33], mentioned above) is 2D free-surface flows with vorticity, although there has been a lot of progress in this direction in the last two decades. This progress has mainly been based on the existence of a scalar stream function, which permits a drastic simplification of the formulation, which is not applicable to 3D problems.

## 2. Differential Formulation of the Problem

*2.1. Generalities, and Description of the Fluid Domain*

In this work, our attention is focused on an inviscid, compressible (barotropic) fluid, undergoing a rotational flow in a horizontally unbounded domain. The fluid domain is limited by a free surface (upper boundary), an impermeable moving bottom (seabed, lower boundary), and—possibly—vertical lateral boundaries, restricting the fluid domain horizontally in some directions (horizontal sectors); see Figure 1. For simplicity, the lateral boundaries, if existing, are assumed to be rigid walls. To give an exact mathematical formulation of this fluid-dynamics problem, an orthogonal Cartesian system $Ox_1x_2z$ is introduced, with $\boldsymbol{x} = (x_1, x_2)$ being the horizontal spatial variables, and $z$ being the vertical variable, pointing upwards, i.e., in the opposite direction with respect to the constant gravity acceleration $\boldsymbol{g}$. The level $z = 0$ is taken to coincide with the quiescent free surface. The moving seabed is located at $z = -h(\boldsymbol{x}, t)$, where $h(\boldsymbol{x}, t)$ is a known depth function of the horizontal variable $\boldsymbol{x}$ and the time $t$, whereas the free surface is described by the equation $z = \eta(\boldsymbol{x}, t)$, where $\eta(\boldsymbol{x}, t)$ is an (unknown) surface-elevation function. Accordingly, the fluid domain is shaped as (see Figure 1):

$$V(t) = \left\{ (\boldsymbol{x}, z) \in \mathbb{R}^3 : \boldsymbol{x} = (x_1, x_2) \in D \subseteq \mathbb{R}^2, z \in [-h(\boldsymbol{x}, t), \eta(\boldsymbol{x}, t)] \right\}, \quad (1)$$

where the horizontal domain $D$ is the projection of the free surface on the plane $(x_1, x_2)$. $D$ is assumed to be unbounded and simply connected.

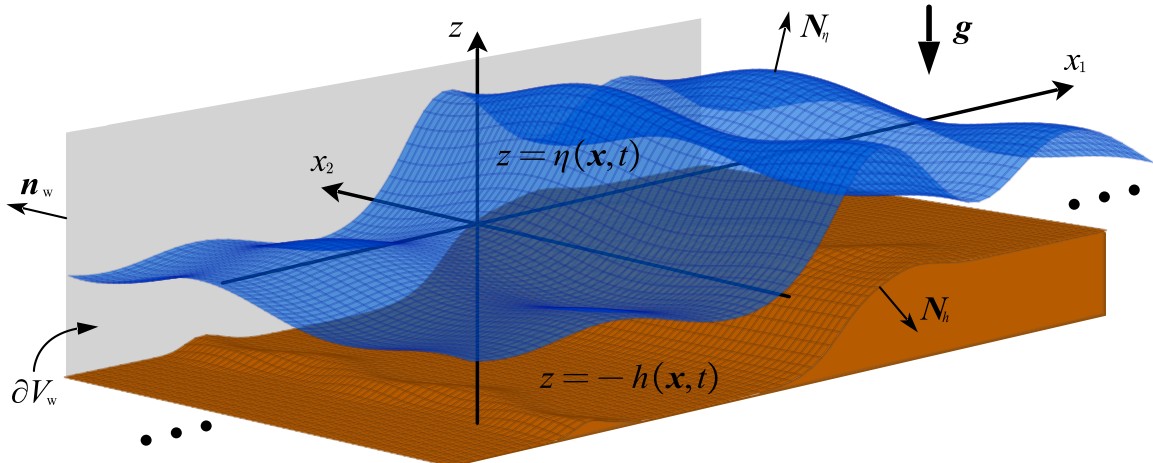

**Figure 1.** Geometrical configuration of the fluid domain $V(t)$.

The vectors:

$$\boldsymbol{N}_\eta \;=\; \nabla \left( z - \eta\left(\boldsymbol{x}, t\right) \right) \;=\; \left( -\frac{\partial\eta}{\partial x_1}, \; -\frac{\partial\eta}{\partial x_2}, \; 1 \right) \tag{2a}$$

and:

$$\boldsymbol{N}_h \;=\; -\nabla \left( z + h\left(\boldsymbol{x}, t\right) \right) \;=\; \left( -\frac{\partial h}{\partial x_1}, \; -\frac{\partial h}{\partial x_2}, \; -1 \right) \tag{2b}$$

are normal (perpendicular) but non-unitary on the free surface and the seabed, respectively, pointing outwardly with respect to the fluid. They commonly appear in our subsequent calculations, and this is why they are given special names.

For convenience, the free surface and the seabed are also denoted by $\partial V_f$ and $\partial V_{sb}$, respectively. The lateral boundary of the fluid domain $V(t)$ is assumed to be vertical, and, as a whole, is denoted by $\partial V_{lat}$. It consists of two types of boundaries: rigid-wall boundaries, denoted by $\partial V_w$, and infinite "boundaries", meaning that the flow extends to infinity in the corresponding horizontal directions, denoted by $\partial V_\infty$. The outward unit normal vector on the lateral boundary is denoted by $\boldsymbol{n}_{lat}$ or $\boldsymbol{n}_w$ when the considerations are restricted on the rigid-wall part.

Of the two descriptions normally considered in hydrodynamics, the Eulerian one is of primary interest to us, although the Lagrangian description will also be touched upon since it is intimately related with Hamilton's variational principle. Although our main goal herein is to provide a variational formulation of rotational free-surface flows, in this section, we shall briefly present the classical differential formulation of the problem. There are good reasons for doing this: (i) Rederiving standard differential equations and boundary conditions from the variational principle is a desirable justification of both approaches; and (ii) some of these equations, especially those expressing the kinematics of the flow, will be used as constraints in the variational principle.

### 2.2. Equations Governing the Bulk Flow

The involved fields in the Eulerian description are: the fluid velocity $\boldsymbol{u} = \boldsymbol{u}\left(\boldsymbol{x}, z, t\right)$, fluid density $\rho = \rho\left(\boldsymbol{x}, z, t\right)$, and fluid pressure $p = p\left(\boldsymbol{x}, z, t\right)$, which, along with the free-surface elevation $\eta = \eta\left(\boldsymbol{x}, t\right)$, constitute the unknowns of the problem. The differential equations, governing the fluid flow in $V(t)$, are the following (see, e.g., [41–43]):

$$\frac{\partial\rho}{\partial t} + \nabla \cdot \left(\rho\,\boldsymbol{u}\right) = 0, \quad \text{(conservation of mass)}, \tag{3}$$

$$\frac{\partial\boldsymbol{u}}{\partial t} + \left(\boldsymbol{u}\cdot\nabla\right)\boldsymbol{u} + \frac{\nabla p}{\rho} = -\boldsymbol{g}, \quad \text{(Euler equation)}, \tag{4}$$

which expresses the momentum law, and the constitutive equation (barotropic fluid):

$$p \;=\; p(\rho) \;=\; \rho^2 \, \frac{\partial E(\rho)}{\partial \rho}, \tag{5}$$

where $E = E(\rho)$ is the internal energy per unit mass of the fluid (defined by phenomeno-logical considerations). Note that, in Equations (3) and (4), $\nabla \cdot \;=\; \left( \frac{\partial \cdot}{\partial x_1}, \frac{\partial \cdot}{\partial x_2}, \frac{\partial \cdot}{\partial z} \right)$ is the 3D gradient operator.

**Remark 2.** In many relevant works, the internal energy is assumed to depend on the density, $\rho$, and the specific entropy, $s$, of the fluid, i.e., $E = E(\rho, s)$, in accordance with general thermodynamic considerations. We do not follow this tradition for various reasons. First, the related necessary assumption of isentropic flow cancels any essential contribution from the introduction of the entropy in the variational analysis, since it produces only irrotational flows; see, e.g., [15] (p. 5). Second, the option of studying non-isentropic flows (which induce heat transfer) based on the classical Hamilton's principle is controversial [14] (p. 345). Third, the introduction of Lin's constraint, expressing the conservation of parcels' identity (see Sections 3.1 and 3.2), resolves the problem of variationally obtaining the equations of rotational flow, initially supposed to be resolved using the specific entropy as an independent field.

### 2.3. Boundary Conditions

Concerning the boundary conditions, we have to distinguish the various types of boundaries. On the free surface, the following two conditions apply:

$$\frac{\partial \eta}{\partial t} \;+\; u_1 \, \frac{\partial \eta}{\partial x_1} \;+\; u_2 \, \frac{\partial \eta}{\partial x_2} \;-\; u_3 \;=\; 0, \text{ on } z \;=\; \eta(\mathbf{x}, t), \tag{6}$$

$$p \;=\; \bar{p} \;=\; \text{given, on } z \;=\; \eta(\mathbf{x}, t), \tag{7}$$

where $\bar{p} = \bar{p}(\mathbf{x}, t)$ is a known applied-pressure field. The first of them is the kinematic condition, ensuring that the free surface moves in accordance with the motion of the fluid parcels lying on it (that is, it is a material surface) while the second one is of a dynamic nature, and it is called the dynamic free-surface condition.

**Remark 3.** In the case of barotropic, irrotational flow, Bernoulli's equation (see, e.g., [41] (Article 20))

$$\frac{p}{\rho} + \frac{\partial \Phi}{\partial t} + \frac{1}{2} u^2 + E + \Omega = C(t)$$

(where $\Phi$ is the velocity potential, $u = \nabla \Phi$, $\Omega = \Omega(x, z)$ is the potential of an external conservative force-field, and $C(t)$ is an arbitrary function of time) permits us to write the dynamic condition (7) as an evolution equation with respect to the velocity potential, greatly facilitating the variational formulation, mathematical study, and numerical treatment of the problem [3,6,8,36,44]. Such an equation is not a priori available for the problem under consideration. However, an extended version of Bernoulli's equation, appropriate for rotational flows, can be derived from the variational formulation presented herein. See also [37,41] and Section 4.4.

On the moving seabed and on the lateral rigid-wall boundaries, we have the kinematic conditions:

$$\frac{\partial h}{\partial t} \;+\; u_1 \, \frac{\partial h}{\partial x_1} \;+\; u_2 \, \frac{\partial h}{\partial x_2} \;+\; u_3 \;=\; 0, \text{ on } z \;=\; -h(\mathbf{x}, t), \tag{8}$$

and:
$$\boldsymbol{u} \cdot \boldsymbol{n}_{\mathrm{w}} = 0, \text{ on } \partial V_{\mathrm{w}}, \tag{9}$$

ensuring zero mass flow through the impermeable boundaries. Recall that, in Equation (9), $\boldsymbol{n}_{\mathrm{w}}$ is the normal unit vector on the rigid-wall boundary $\partial V_{\mathrm{w}}$. Since the motion of these boundaries is predetermined, a second (dynamical) condition is not needed.

Finally, concerning the lateral "infinite" boundary $\partial V_{\infty}$, that is, the flow at infinity in the unbounded horizontal directions, we assume that the velocity field and the free-surface elevation tend to zero with rates ensuring that the energy integral is finite. In the opposite case, when a wave system of infinite energy has been developed and moves towards infinity, the variational method is not directly applicable. Nevertheless, such cases can be included in the present formulation using domain decomposition techniques. Such kinds of techniques have been well developed in the context of irrotational free-surface flows; see, e.g., [45–47].

### 2.4. A Brief Discussion of the Lagrangian Approach

Analytical mechanics is based on the consideration of material particles' positions $\boldsymbol{x}_i(t)$ and their virtual variations $\delta \boldsymbol{x}_i(t)$, subjected to all relevant constraints. For a continuum, especially a fluid, the index identifying the positions of material parcels is a vector field, leading to an expression of the form $\boldsymbol{x}_{\boldsymbol{a}}(t) = \boldsymbol{x}(\boldsymbol{a}, t)$. The common choice of index (label) $\boldsymbol{a}$ is the initial position of the fluid parcels, which means that $\boldsymbol{x}(\boldsymbol{a}, t_0) = \boldsymbol{a}$. Clearly, in this consideration, $\boldsymbol{a}$ is an independent variable extending over the geometric domain $V(t_0)$, not varying with time. For each $t \geq t_0$, the transformation:

$$V(t_0) \ni \boldsymbol{a} \to \boldsymbol{x}(\boldsymbol{a}, t) \in V(t) \tag{10a}$$

is assumed to be regular, smooth, and invertible (diffeomorphism). Thus, the inverse transformation:

$$V(t) \ni \boldsymbol{x} \to \boldsymbol{a}(\boldsymbol{x}, t) \in V(t_0), \tag{10b}$$

is well defined, giving $\boldsymbol{a}$ a representation as a function of $(\boldsymbol{x}, t)$. This means that to a fluid parcel lying at the position $\boldsymbol{x}$ in the time instant $t$, it corresponds to an initial position $\boldsymbol{a} = \boldsymbol{a}(\boldsymbol{x}, t)$. Accordingly, the same $\boldsymbol{a}-$value is assigned to all $(\boldsymbol{x}, t)$ shaping a fluid-parcel trajectory. This means that the field $\boldsymbol{a}(\boldsymbol{x}, t)$ is invariant along trajectories, a fact equivalent to the equation:

$$\frac{D \boldsymbol{a}(\boldsymbol{x}, z, t)}{D t} \equiv \frac{\partial \boldsymbol{a}}{\partial t} + (\boldsymbol{u} \cdot \nabla) \boldsymbol{a} = 0. \tag{11}$$

Equation (11), usually referred to as the *conservation of identity*, will be used as a kinematical constraint in the Hamiltonian action functional. Equations for the conservation of mass and the conservation of momentum in terms of the parcel-position field $\boldsymbol{x}(\boldsymbol{a}, t)$ can be found in many books of hydrodynamics (see, e.g., [41,48]). They are not presented herein because they are not needed in our study.

**Remark 4.** All the fields involved in our considerations, both geometrical, as $\eta(\boldsymbol{x}, t)$ and $h(\boldsymbol{x}, t)$, and physical, as $\rho(\boldsymbol{x}, z, t)$, $\boldsymbol{u}(\boldsymbol{x}, z, t)$, etc., are considered to be sufficiently smooth. In particular, for most of our analysis, $C^1-$smoothness suffices.

## 3. Variational Formulation of the Problem
### 3.1. Preliminary Remarks, and The Variational Controversy

As it is well known, the dynamical equations of any ideal (non-dissipative) mechanical system, subjected to external loads, can be obtained by Hamilton's principle in the form:

$$\delta \int_T \mathcal{L} Dt + \int_T \delta W dt = 0 \tag{12}$$

where $\mathcal{L}$ is the Lagrangian function of the system, $\delta W$ is the virtual work of the loads not taken into account by the Lagrangian, and $T = [t_0, t_1]$ is an arbitrary time interval. Equation (12) is easily obtained by means of the principle of virtual work [49,50]. A quick direct proof is given in [51], Section 4.2.1. The fundamental (primitive) Lagrangian function is the kinetic minus potential energy, augmented by appropriate, case-dependent, constraints. The fluid-flow problem examined herein belongs to the class of problems governed by Equation (12), from which we should be able to derive all the governing equations and boundary conditions, with the proviso that it is correctly implemented. The latter statement means that the variations have to be performed in accordance with the assumptions underlying Hamilton's principle.

In the Eulerian description of fluid flow, Equations (3)–(9), the fluid-parcel positions $X(a, t)$ do not come into play and, consequently, the information concerning the parcels' individuality and virtual variations of their positions is apparently lost. This fact is the source of the *variational controversy* of Eulerian fluid dynamics, which has also been discussed in the Introduction. This variational controversy has been solved, at least for the bulk motion of the fluid inside $V(t)$, by means of the clever proposal of Lin [16,17] to introduce Equation (11), along with the conservation of mass, Equation (3), as constraints in the Eulerian action functional. This miraculous trick, whose physical interpretation needs further clarification (at least, to the authors' opinion), results in an augmented action functional (see Equation (16), in Section 3.2) that contains both Eulerian and Lagrangian elements. Such a mixed formulation seems to be unpleasant and difficult to use, at first glance. However, this difficulty is greatly mitigated since $a(x, z, t)$, and the corresponding Lagrange multiplier $A(x, z, t)$ gain a new role through the Euler–Lagrange equations of the variational principle as elements of a representation of the velocity and pressure fields in their Eulerian formulation; see Equations (32) and (33). These representations are generalizations of similar ones given by Clebsch [37] for incompressible flows. The variational reconstruction of the Clebsch representation has been reconsidered by Bateman [38,39] for 2D compressible flows and has been derived for general 3D flows by Serrin [16]. This issue is discussed in detail by Seliger and Whitham in their seminal paper [15]; see also [52].

Nice as this phenomenon might be, it seems that it does not resolve the variational controversy on the boundaries of the fluid domain. To resolve this problem, we introduce corresponding *differential-variational constraints*, which allow us to express the Eulerian variations on the boundaries in terms of the Lagrangian ones, resulting in correct kinematic and dynamic boundary conditions. This is the main original contribution of the present work, which is further developed in Sections 3.3, 3.4 and 5.

### 3.2. The Hamiltonian Action Functional

Since the fluid-flow problem examined herein refers to an ideal mechanical system, its dynamics must be governed by Hamilton's principle. (As always in this paper, we principally refer to the Eulerian formalism of hydrodynamics). This fact was recognized in the early 20th century (see, e.g., [13,14]), and it has been the starting point of many relevant investigations since then. According to this line of thought, the basic (primitive) Lagrangian function is defined as the kinetic minus potential energy of the fluid, that is:

$$\mathcal{L}_{prim} = \int_{V(t)} \rho \left( \frac{u^2}{2} - E(\rho) - P \right) dV, \tag{13}$$

where $P = P(x, z)$ is the gravity-force potential. Following Herivel, Lin, Serrin, and other authors [14–17,19,22], the constraints of the mass conservation, Equation (3), and identity conservation, Equation (11), are also introduced in Equation (13), obtaining the augmented Lagrangian:

$$\mathcal{L} = \int_{V(t)} \left\{ \rho \left( \frac{u^2}{2} - E(\rho) - P \right) - k \left( \frac{\partial \rho}{\partial t} + \nabla \cdot (\rho \, u) \right) - \rho \, A \, \frac{D \, a}{d \, t} \right\} dV, \tag{14}$$

where $k = k(\boldsymbol{x}, z, t)$ and $\boldsymbol{A} = \boldsymbol{A}(\boldsymbol{x}, z, t) = (A_1, A_2, A_3)$ are appropriate Lagrange multipliers.

What remains to completely specify a variational principle, Equation (12), for our fluid-flow problem is the virtual work of the external loads acting on the boundaries of the fluid domain. It turns out that it is sufficient to specify only the virtual work of loads acting on those parts of the boundary for which a dynamic boundary condition is applied. Since rigid-wall lateral boundaries are assumed to be fixed, and since the seabed may undergo only predetermined (known) motion, only the free surface belongs to this category. The external load on the free surface is realized by means of an applied pressure $\overline{p}(\boldsymbol{x}, t)$, and the corresponding virtual work is given by the formula:

$$\delta W|_{\partial V_f} = -\int_D \overline{p}\ \delta\eta\ d x_1\ d x_2 = -\delta \int_D \overline{p}\ \eta\ d x_1\ d x_2, \tag{15}$$

where $\delta\eta$ is the virtual variation of the free-surface elevation. Note that Equation (15) is exactly the same as in the irrotational case.

Combining the Lagrangian function (14) and the virtual work (15) we are now in a position to state the complete, augmented action functional for our problem:

$$
\begin{aligned}
&\widetilde{\mathscr{S}}\,[\,\boldsymbol{a},\,\boldsymbol{u},\,\rho,\,\boldsymbol{A},\,k,\,\eta\,]\\
&= \int_T \int_D \int_{-h(\boldsymbol{x},t)}^{\eta(\boldsymbol{x},t)} \left\{ \rho\,(\cdots)\,\left(\tfrac{u^2}{2}(\cdots) - E(\rho(\cdots)) - P(\boldsymbol{x}, z)\right)\right.\\
&\quad \left. - k(\cdots)\left(\tfrac{\partial\rho(\cdots)}{\partial t} + \nabla\cdot(\rho(\cdots)\,\boldsymbol{u}(\cdots))\right) - \rho(\cdots)\,\boldsymbol{A}(\cdots)\,\tfrac{d\,\boldsymbol{a}(\cdots)}{dt}\right\}\,dz\,d\boldsymbol{x}\,dt\\
&\quad - \int_T \int_D \overline{p}(\boldsymbol{x}, t)\,\eta(\boldsymbol{x}, t)\,d\boldsymbol{x}\,dt,
\end{aligned}
\tag{16}
$$

where $(\cdots)$ stands for $(\boldsymbol{x}, z, t)$, and, as in Equation (12), and $\mathrm{T} = [\,t_0,\,t_1\,]$ is an arbitrary time interval. Then, the global variational equation, which is expected to govern the dynamics of the studied problem, takes the form:

$$\delta\,\widetilde{\mathscr{S}}\,[\,\boldsymbol{a},\,\boldsymbol{u},\,\rho,\,\boldsymbol{A},\,k,\,\eta\,] = 0. \tag{17}$$

### 3.3. Differential-Variational Constraints and Boundary Virtual Displacements

Although Lin's constraint resolved the variational controversy within the 3D fluid domain, permitting us to consider the Eulerian variations $\delta\rho(\boldsymbol{x}, z, t)$, $\delta\boldsymbol{u}(\boldsymbol{x}, z, t)$, etc. as independent within $V(t)$, it seems that the problem remains unsolved on the boundary of the fluid domain; see Section 5. Conceptual a priori arguments supporting this statement are the following:

(i)   Lin's constraint is of integral character, acting within the 3D fluid domain $V(t)$. Thus, there is no strong reason to believe that such a constraint works equally well on the boundary, which is a lower-dimension manifold.

(ii)  There is no evidence that Lin's constraint can handle the variational controversy on moving boundaries, such as the free surface and the moving seabed.

Nevertheless, the physics introduced via the two considered constraints seem to be appropriate and adequate to solve the problem. To cope with the difficulties on the boundary, we persist in considering the same constraints, exploiting now localized versions of them, which lead to differential-variational relations. The latter, being valid pointwise, are able to take into account the 2D nature of the boundaries and their motion. Specifically, the differential-variational relations associate Lagrangian variations (for fixed $\boldsymbol{a}$, $t$) with Eulerian ones (for fixed $\boldsymbol{x}$, $z$, $t$) for the same field written in Eulerian variables. To obtain these relations, we need to consider both the Lagrangian and Eulerian descriptions of the fluid flow, and introduce two variation operators: $\delta_L(\cdot)$ for variations corresponding to fixed $(\boldsymbol{a}, t)$, and $\delta(\cdot)$ for variations corresponding to fixed $(\boldsymbol{x}, z, t)$. We further recall that in the Lagrangian description of the flow, parcel trajectories are defined by equations of the

form $X = X(a, t) = (X_1, X_2, Z)(a, t)$ while, in the Eulerian description, labels $a$ can be considered as a spatial field, $a = a(x, z, t)$, in the sense discussed in Section 2.4. This said, we are in a position to state the relation between the two variation operators, $\delta_L(\cdot)$ and $\delta(\cdot)$:

$$\delta_L(\cdot) = \delta(\cdot) + (\delta_L X \cdot \nabla)(\cdot), \tag{18}$$

for any field of the fluid written in Eulerian independent variables $x$, $z$, $t$. In the above Equation (18), $\delta_L X$ is the virtual displacement of the fluid parcels, also written in Eulerian variables, that is, $\delta_L X = \delta_L X(a(x, z, t), t)$. An analytical proof of Equation (18) is given by Gelfand and Fomin [53] (Section 37), in the abstract setting of Calculus of Variations, without direct mention to Lagrangian and Eulerian hydrodynamics. A schematic illustration of the main idea behind the derivation is presented in [21]. A graphical derivation of Equation (18), inspired by Gelfand and Fomin's analysis, can also be found in [51] (Section 6.6). The latter is somewhat obscure, using the natural time of the system also as an artificial "time" associated with the variations. See also [54] (p. 18).

To obtain the differential-variational version of the mass conservation, we apply Equation (18) to $\rho$, $\delta_L \rho = \delta \rho + \delta_L X \cdot \nabla \rho$, and combine the latter with the relation $\delta_L \rho = -\rho(\nabla \cdot \delta_L X)$, which occurs from the mass conservation in the Lagrangian setting; see Equation (6) of Bretherton [21] (Section 2). Thus, eliminating $\delta_L \rho$, we obtain:

$$\delta \rho = -\nabla \cdot (\rho \, \delta_L X). \tag{19a}$$

The differential-variational form of the parcels' identity conservation is easily derived by applying Equation (18) to the labels $a$ and combining it with the obvious identity $\delta_L a = 0$. The result is the formula:

$$\delta a = -(\delta_L X \cdot \nabla) a. \tag{19b}$$

Equations (19a) and (19b) relate the Eulerian variations $\delta \rho$ and $\delta a$ with the virtual displacements $\delta_L X$ of the fluid parcels. These results will be exploited for the treatment of the boundary-integral terms of the variation of the functional (16). In these boundary terms, apart from $\delta \rho$ and $\delta a$, the variation of the velocity field, $\delta u$, also appears. This fact motivates the derivation of an expression between $\delta u$ and $\delta_L X$ as well. Since $u = D X / d t$ in the Lagrangian description, we have that $\delta_L u = \delta_L(D X / D t)$. Introducing the latter into Equation (18) applied to $u$, we find:

$$\delta u = \frac{D}{D t}(\delta_L X) - (\delta_L X \cdot \nabla) u. \tag{19c}$$

Equations (19) are essential for the subsequent analysis of the boundary terms of the variational Equation (17); see Section 5. Thus, it is worthwhile to notice the following features:

- Equations (19) are point-wise conditions applying to any fluid parcel, lying either in the interior or on the boundary of the fluid domain $V(t)$.
- Using Equations (19), the variational equation of the action functional can be re-expressed in terms only of $\delta_L X$.
- Equations (19), when applied to boundary points, should be combined with any additional constraints on the virtual displacements $\delta_L X$, implied by the geometry and the motion of the boundaries.

The last point needs some more elaboration. First, we introduce the notation $\delta_L X_\eta$, $\delta_L X_h$, and $\delta_L X_w$ for the virtual displacements of the fluid parcels lying on the free surface, seabed, and lateral rigid-wall boundary, respectively. The synoptic notation $\delta_L X_b$, $b \in \{\eta, h, w\}$, will also be used, to facilitate general statements concerning boundary virtual displacements. Then, we observe that in $\delta_L X_b = \delta_L X_b(x, z, t)$, the arguments $(x, z, t)$ are not independent but should, instead, satisfy the equations defining the corresponding boundary. For example:

$S_\eta \ (\pmb{x}, \ z, \ t) \ \equiv \ z \ - \ \eta \ (\pmb{x}, \ t) \ = 0$ on the free surface; and
$S_h \ (\pmb{x}, \ z, \ t) \ \equiv \ - \, z \ - \ h \ (\pmb{x}, \ t) \ = \ 0$ on the seabed.

Finally, we state the following conditions for $\delta_L \ \pmb{X}_b$, dictated by the geometry and kinematics of the corresponding boundaries, in conjunction with the defining properties of the virtual displacements:

On the free surface, $\partial \, V_f$, we have that:

$$\delta_L \, \pmb{X}_\eta \ \cdot \ \pmb{N}_\eta \ = \ \delta \, \eta, \ (\pmb{x}, \ z, \ t) \ \in \ \partial \, V_f \, (t). \tag{20}$$

Equation (20) is proved as follows. Since $\partial \, V_f$ is a freely moving, unknown boundary, $\delta_L \, \pmb{X}_\eta$ are arbitrary on it. However, $\partial \, V_f$ is defined by the unknown surface-elevation field $\eta \ = \ \eta \ (\pmb{x}, \ t)$ and, thus, its variation $\delta \, \eta$ must be related with $\delta_L \, \pmb{X}_\eta$. Since the free surface is a material surface, Equation (18) leads to $0 \ = \ \delta_L \, S_\eta \ = \ \delta \, S_\eta \ + \ \delta_L \, \pmb{X}_\eta \ \cdot \ \nabla \, S_\eta$. From this equation, along with the relations $\delta \, S_\eta \ = \ - \, \delta \, \eta$ and $\nabla \, S_\eta \ = \ \pmb{N}_\eta$ (see also Equation (2a)), we obtain Equation (20).

Working similarly, we find the following equation for the virtual displacement on the moving seabed:

$$\delta_L \, \pmb{X}_h \ \cdot \ \pmb{N}_h \ = \ 0, \ (\pmb{x}, z, \ t) \ \in \ \partial \, V_{s \, b} \, (t), \tag{21}$$

since the motion of this part of the boundary is predetermined. Here, $\pmb{N}_h \ = \ \nabla \, S_h$ (see, also, Equation (2b)).

Last, on the lateral rigid-wall boundary, $\partial \, V_w$, the variation should be compatible with the impermeability condition, which leads to:

$$\delta_L \, \pmb{X}_w \ \cdot \ \pmb{n}_w \ = \ 0, \ (\pmb{x}, z) \ \in \ \partial \, V_w. \tag{22}$$

*3.4. Statement of the Variational Principle and Outline of Its Proof*

Having introduced all the necessary concepts and notations, we are now able to state the variational principle governing the problem under study in the form of a theorem:

**Theorem 1.** If the 12 scalar fields $\pmb{a}$, $\pmb{A}$, $\pmb{u}$, $k$, $\rho$, $\eta$ render the augmented Hamiltonian action functional, Equation (16), stationary for any set of variations:

- $\delta \, \pmb{a}$, $\delta \pmb{A}$, $\delta \, \pmb{u}$, $\delta \, k$, $\delta \, \rho$ within the fluid volume $V \, (t)$, where they are all considered as independent of each other; and
- $\delta \, \pmb{a}$, $\delta \, \pmb{u}$, $\delta \, \rho$, $\delta \, \eta$ on the boundary $\partial \, V \, (t)$ of the fluid (the variations $\delta \pmb{A}$, $\delta \, k$ do not appear on the boundary terms), where they are restricted by means of Equations (19) and (20),

then, the fields $\rho, \pmb{u} \ = \ - \ \nabla \, k \ + \ \pmb{A} \, \nabla \, \pmb{a}$ and:

$$p \ = \ \rho \left( \frac{\partial \, k}{\partial \, t} \ - \ \pmb{A} \, \frac{\partial \, \pmb{a}}{\partial \, t} \ - \ \frac{\pmb{u}^2}{2} \ - \ E \ - \ P \right),$$

satisfy the differential Equations (3) and (4) within $V \, (t)$, and the boundary conditions (6), (7) on the free surface, (8) moving seabed, and (9) lateral rigid-wall. ■

The proof of this theorem is lengthy, and it is given in Sections 4 and 5. To help the reader follow the laborious details, the proof's three main steps are outlined as follows:

**Step 1:** First, in Section 4.1, we calculate the partial Gateaux derivatives:

$$\delta_q \, \widetilde{\mathscr{S}} \, [\, \pmb{a}, \ \rho, \ \pmb{A}, \ k, \ \pmb{u}, \ \eta \, ; \ \delta \, q \,], \quad q \ \in \ \{ \, \pmb{a}, \ \rho, \ \pmb{A}, \ k, \ \pmb{u}, \ \eta \ \}, \tag{23}$$

of the augmented action functional (16), and write the global variational Equation (17) in the expanded form:

$$
\sum_q \iint_{TD-} \int_{h(x,t)}^{\eta(x,t)} (\cdots)\, \delta q \, dz \, dx \, dt + \sum_q \iint_{TD} [\,(\cdots)\delta q\,]_{z=\eta}\, dx \, dt
$$

$$
\underbrace{\phantom{\sum_q \iint_{TD-} \int_{h(x,t)}^{\eta(x,t)} (\cdots)\, \delta q \, dz \, dx \, dt}}_{\text{volume integral terms}} + \underbrace{\phantom{\sum_q \iint_{TD} [\,(\cdots)\delta q\,]_{z=\eta}\, dx \, dt}}_{\text{free-surface boundary terms}}
$$

$$
+ \underbrace{\sum_q \iint_{TD} [\,(\cdots)\delta q\,]_{z=-h}\, dx \, dt}_{\text{seabed boundary terms}} + \underbrace{\sum_q \int_T \int_{\partial V_w} (\cdots)\, \delta q \, dS_w \, dt}_{\text{lateral rigid-wall terms}} = 0. \tag{24}
$$

In the last sum of (boundary) integrals, appearing in the left-hand side of Equation (24), the integration is taken over the whole lateral boundary. However, it suffices to keep the integral only on the rigid-wall part $\partial V_{\mathrm{w}}$, as shown above, since the variations of the flow fields are taken to vanish on the "infinite" lateral boundary $\partial V_\infty$.

**Step 2:** Then, we consider variations that vanish on the boundaries, and obtain the individual variational equations (since $\delta q$ are considered independent):

$$
\iint_{TD-} \int_{h(x,t)}^{\eta(x,t)} (\cdots)\, \delta q \, dz \, dx \, dt = 0, \quad q \in \{\, a, \rho, A, k, u \,\}, \tag{25}
$$

corresponding to the volume integral terms (what we call "volume integral terms" and "boundary integral terms" here are, in fact, time-volume and time-boundary integrals). These equations, in conjunction with the fact that $\delta q$ are arbitrary within the fluid domain, provide us with five Euler–Lagrange equations (see Section 4.2), which are further discussed in Sections 4.3 and 4.4. The equations obtained in this step are well known, so that the first two steps do not produce original results. They are, however, necessary prerequisites for the next step of the proof, where the original contribution of this paper lies.

**Step 3:** This is taken in Section 5. Substituting the Euler–Lagrange equations into the global variational Equation (24), the volume integral terms are eliminated and only the boundary integral terms remain, associated with the free surface, seabed, and lateral boundaries. As discussed in Section 3.3, to resolve the variational controversy on the boundary, the differential-variational constraints Equations (19) and (20) are additionally imposed on the boundary variations of the involved Eulerian fields; namely, $\delta \rho$, $\delta a$, $\delta u$, and $\delta \eta$. Thus, the latter cannot be considered as independent, but, observing Equations (19) and (20), they may all be expressed in terms of the boundary parcels' virtual displacements $\delta_L X_b = \delta_L X_b (x, z, t)$. Accordingly, the global variational Equation (24) reduces to one restricted on the boundary of the fluid domain, involving only the variations $\delta_L X_\eta$, $\delta_L X_h$, $\delta_L X_{\mathrm{w}}$. The latter equation, in conjunction with Equations (21) and (22), and the standard variational arguments provide us with all (kinematic and dynamic) boundary conditions for the three kinds of boundaries of the studied problem. To the best of our knowledge, this set of boundary conditions is variationally derived for the first time.

## 4. Calculation of Variations and Euler–Lagrange Equations within the Fluid Domain

### 4.1. Partial Gateaux Derivatives of the Action Functional

Now, we proceed with the implementation of the methodology outlined in Section 3.4 in order to prove Theorem 1. The starting point is the calculation of the partial Gateaux derivatives of the augmented action functional, Equation (16), with respect to all its arguments.

The Gateaux derivatives with respect to the Lagrange multipliers $k$ and $A$ are trivially calculated, resulting in:

$$
\delta_k \widetilde{\mathscr{S}} = - \iint_{TD-} \int_{h(x,t)}^{\eta(x,t)} \left( \frac{\partial \rho}{\partial t} + \nabla \cdot (\rho\, u) \right) \delta k \, dz \, dx \, dt, \tag{26a}
$$

and:

$$\delta_A \widetilde{\mathscr{F}} \; = \; - \iint_{T\, D-} \int_{h\,(\boldsymbol{x},\,t)}^{\eta\,(\boldsymbol{x},\,t)} \rho \; \frac{D\,\boldsymbol{a}}{D\,t} \; \delta A \; d\,z \; d\,\boldsymbol{x} \; d\,t. \tag{26b}$$

The three Gateaux derivatives, with respect to the fields $\boldsymbol{a}$, $A$, and $\boldsymbol{u}$, are much more involved, requiring extensive calculations. The corresponding results, in a form appropriate for the subsequent variational analysis, are as follows:

$$
\begin{aligned}
\delta_a \widetilde{\mathscr{F}} \; = \; & \int_T \int_D \int_{-\,h\,(\boldsymbol{x},\,t)}^{\eta\,(\boldsymbol{x},\,t)} \left( \frac{D\,(\rho\,A)}{D\,t} \; + \; (\nabla \; \cdot \; \boldsymbol{u})\,\rho\,A \right) \; \delta\,\boldsymbol{a} \; d\,z \; d\,\boldsymbol{x} \; d\,t \\
& + \int_T \int_D \left( \frac{\partial\,\eta}{\partial\,t} \; - \; [\,\boldsymbol{u}\,]_{z\,=\,\eta} \; \mathbf{N}_\eta \right) \; [\,\rho\,A\,\delta\,\boldsymbol{a}\,]_{z\,=\,\eta} \; d\,\boldsymbol{x} \; d\,t \\
& + \int_T \int_D \left( \frac{\partial\,h}{\partial\,t} \; - \; [\,\boldsymbol{u}\,]_{z\,=\,-\,h} \; \mathbf{N}_h \right) \; [\,\rho\,A\,\delta\,\boldsymbol{a}\,]_{z\,=\,-\,h} \; d\,\boldsymbol{x} \; d\,t \\
& - \int_T \int_{\partial\,D_w} \int_{-\,h\,(\boldsymbol{x},\,t)}^{\eta\,(\boldsymbol{x},\,t)} (\rho\,A\,\delta\,\boldsymbol{a})\,\boldsymbol{u}\,\boldsymbol{n}_w \; d\,z \; d\,l \; d\,t \,,
\end{aligned}
\tag{26c}
$$

$$
\begin{aligned}
\delta_\rho \widetilde{\mathscr{F}} \; = \; & \int_T \int_D \int_{-\,h\,(\boldsymbol{x},\,t)}^{\eta\,(\boldsymbol{x},\,t)} \left( \frac{u^2}{2} \; - \; E \; - \; \rho\,\frac{\partial E}{\partial \rho} \; - \; P \; + \; \frac{D\,k}{D\,t} \; - \; A\,\frac{D\,a}{D\,t} \right) \; \delta\,\rho \; d\,z \; d\,\boldsymbol{x} \; d\,t \\
& + \int_T \int_D \left( \frac{\partial\,\eta}{\partial\,t} \; - \; [\,\boldsymbol{u}\,]_{z\,=\,\eta} \; \mathbf{N}_\eta \right) \; [\,k\,\delta\,\rho\,]_{z\,=\,\eta} \; d\,\boldsymbol{x} \; d\,t \\
& + \int_T \int_D \left( \frac{\partial\,h}{\partial\,t} \; - \; [\,\boldsymbol{u}\,]_{z\,=\,-\,h} \; \mathbf{N}_h \right) \; [\,k\,\delta\,\rho\,]_{z\,=\,-\,h} \; d\,\boldsymbol{x} \; d\,t \\
& - \int_T \int_{\partial\,D_w} \int_{-\,h\,(\boldsymbol{x},\,t)}^{\eta\,(\boldsymbol{x},\,t)} (k\,\delta\,\rho)\,\boldsymbol{u}\,\boldsymbol{n}_w \; d\,z \; d\,l \; d\,t
\end{aligned}
\tag{26d}
$$

and:

$$
\begin{aligned}
\delta_u \widetilde{\mathscr{F}} \; = \; & \int_T \int_D \int_{-\,h\,(\boldsymbol{x},\,t)}^{\eta\,(\boldsymbol{x},\,t)} \rho\,(\boldsymbol{u} \; + \; \nabla\,k \; - \; A\,\nabla\,a) \; \delta\,\boldsymbol{u} \; d\,z \; d\,\boldsymbol{x} \; d\,t \\
& - \int_T \int_D \left[\,\rho\,k\,\delta\,\boldsymbol{u}\,\mathbf{N}_\eta\,\right]_{z\,=\,\eta} \; d\,\boldsymbol{x} \; d\,t \\
& - \int_T \int_D \left[\,\rho\,k\,\delta\,\boldsymbol{u}\,\mathbf{N}_h\,\right]_{z\,=\,-\,h} \; d\,\boldsymbol{x} \; d\,t \\
& - \int_T \int_{\partial\,D_w} \int_{-\,h\,(\boldsymbol{x},\,t)}^{\eta\,(\boldsymbol{x},\,t)} \rho\,k\,\delta\,\boldsymbol{u}\,\boldsymbol{n}_w \; d\,z \; d\,l \; d\,t \,,
\end{aligned}
\tag{26e}
$$

where (we recall that) $D$ is the projection of the free surface on the horizontal plane, $\boldsymbol{n}_w$ is the outward unit normal vector on the (vertical) rigid wall, and $\mathbf{N}_\eta$, $\mathbf{N}_h$ are defined by Equations (2). Further, $\partial\,D_w$ is the intersection of the lateral rigid-wall boundary with $D$ (that is, $\partial\,D_w$ is a line), and $D\,l$ is the corresponding line(detailed derivations of Equations (26c)–(26e) are presented in Appendix A). The first integral in the right-hand side of each of Equations (26c)–(26e) is a volume integral, over the whole fluid domain $V\,(t)$, while the second, third, and fourth integrals are surface integrals taken over the free surface, seabed, and te lateral rigid wall, respectively.

It remains to calculate the Gateaux derivative with respect to the free-surface elevation $\eta$. This is easily performed by invoking the Leibnitz integral rule, resulting in:

$$\delta_\eta \widetilde{\mathscr{F}} = \int_T \int_D \left[ -\,\overline{p} \; + \; \rho\,\left( \frac{1}{2}\,u^2 \; - \; E \; - \; P \right) \; - \; k\,\left( \frac{\partial\,\rho}{\partial\,t} \; + \; \nabla \; \cdot \; (\rho\,\boldsymbol{u}) \right) \; - \; \rho\,A\,\frac{D\,a}{D\,t} \right]_{z\,=\,\eta} \delta\,\eta\,d\,\boldsymbol{x}\,d\,t. \tag{26f}$$

Consequently, the total variation of the action functional (16) takes the form:

$$\delta \widetilde{\mathscr{S}} \; = \; \sum_q \delta_q \widetilde{\mathscr{S}}, \, q \; \in \; \{\, \boldsymbol{a}, \, \rho, \, \boldsymbol{A}, \, k, \, \boldsymbol{u}, \, \eta \,\}, \tag{27}$$

where $\delta_q \widetilde{\mathscr{S}}$ are given by Equations (26a)–(26f).

### 4.2. Euler–Lagrange Equations Corresponding to Variations within the Fluid Domain

Having calculated the partial Gateaux derivatives of the action functional, Equation (26), we are able to proceed with the second step of Section 3.4, considering variations $\delta \boldsymbol{a}$, $\delta \rho$, $\delta \boldsymbol{A}$, $\delta k$, $\delta \boldsymbol{u}$ that vanish on the boundary of the fluid domain $V(t)$, and $\delta \eta = 0$. Thus, only volume integral terms survive in the global variational equation. Now, recalling that the variations $\delta \boldsymbol{a}(\boldsymbol{x}, z, t)$, $\delta \rho(\boldsymbol{x}, z, t)$, etc., $(\boldsymbol{x}, z) \in V(t)$ can be considered as independent of each other and arbitrary, and using the standard argument of the calculus of variations, we obtain:

$$\delta k : \tfrac{\partial \rho}{\partial t} + \nabla \cdot (\rho \, \boldsymbol{u}) = 0 \quad , (\boldsymbol{x}, z) \in V(t), \tag{28}$$

$$\delta \boldsymbol{A} : \frac{D \, \boldsymbol{a}}{D \, t} \; \equiv \; \frac{\partial \, \boldsymbol{a}}{\partial \, t} + (\boldsymbol{u} \, \cdot \, \nabla) \, \boldsymbol{a} \; = \; 0, \, (\boldsymbol{x}, z) \in \, V(t), \tag{29}$$

$$\delta \boldsymbol{a} : \quad \frac{D \, (\rho \, \boldsymbol{A})}{D \, t} + (\boldsymbol{u} \, \cdot \, \nabla) \, (\rho \, \boldsymbol{A}) \; = \; 0$$

which, using Equation (28), is rewritten as:

$$\frac{D \, \boldsymbol{A}}{D \, t} \; \equiv \; \frac{\partial \, \boldsymbol{A}}{\partial \, t} + (\boldsymbol{u} \, \cdot \, \nabla) \, \boldsymbol{A} \; = \; 0, \, (\boldsymbol{x}, z) \in \, V(t), \tag{30}$$

$$\delta \rho : \frac{D \, k}{D \, t} = -\frac{\boldsymbol{u}^2}{2} + E + \rho \, \frac{\partial E}{\partial \rho} + P + \boldsymbol{A} \, \frac{D \, \boldsymbol{a}}{D \, t}, \, (\boldsymbol{x}, z) \in \, V(t), \tag{31}$$

$$\delta \boldsymbol{u} : \boldsymbol{u} = -\nabla k + \boldsymbol{A} \, \nabla \, \boldsymbol{a} \; = \; -\nabla k + \sum_{i=1}^{3} A_i \, \nabla \, a_i, \, (\boldsymbol{x}, z) \in \, V(t). \tag{32}$$

Equations (28) and (29) express the conservation of mass and the conservation of fluid parcels' identity. Equation (30) shows that the Lagrange multipliers $\boldsymbol{A}$ are integrals (constants) of the fluid motion. Equation (31), simplified by setting its last term equal to zero (because of Equation (29)), provides an evolution equation for the Lagrange multiplier $k$. Finally, Equation (32) defines a representation of the velocity field in terms of the fields $k$, $\boldsymbol{a}$, $\boldsymbol{A}$. These equations have been obtained (sometimes with a slightly different form) by many authors since 1959, first appearing in [16]. Euler's momentum equation is not directly included in the above list; however, it can be derived by combining Equations (29)–(32) and eliminating the fields $k$, $\boldsymbol{a}$, $\boldsymbol{A}$, which do not appear in the standard Eulerian formalism. The detailed derivation is given in various books or papers, e.g., [16,19,52,55], and, thus, it is not repeated here. Inverting this point of view, we may say that, in the present variational formulation, Euler's momentum Equation (4) is decomposed into the system of Equations (29)–(32). In some sense, Equations (28)–(32) provide us with a new formulation of the rotational flow problem. This formulation is Eulerian in the sense that all the involved fields depend on the Eulerian variables $(\boldsymbol{x}, z)$, but it also contains the additional fields $k$, $\boldsymbol{a}$, $\boldsymbol{A}$, introduced for resolving the variational controversy. The origin of the latter fields is traced back to the Lagrangian description.

### 4.3. On the Representation of the Fluid Velocity by Means of Potentials

Equation (32) gives to the three (seven scalar) fields $k$, $\boldsymbol{a}$, $\boldsymbol{A}$ the role of extended potentials, defining a representation of the velocity field. It is not clear, however, how many pairs $A_i$, $a_i$ should be used in the term $\boldsymbol{A} \, \nabla \, \boldsymbol{a} = \sum_i A_i \, \nabla \, a_i$. Although, in our formulation, we consider $\boldsymbol{a}$ and $\boldsymbol{A}$ as 3D fields, everything can be repeated successfully

by assuming that *a* and *A* are scalar fields), obtaining, then, a simpler representation of the velocity by means of three scalar potentials. The latter is exactly the same as the representation derived by Clebsch in [37], using different arguments. The question of how many terms should be kept in the representation $u = -\nabla k + \sum A_i \nabla a_i$ is interesting and important, and has been discussed by many authors; see, e.g., [21] (Section 6) [22,26,27,31] (Section 9.3). The simple choice of Clebsch representation seems reasonable (since it involves only three scalar potentials) but leads to restrictions on the structure of the vorticity field. In fact, there exist several works pointing out that the use of the classical Clebsch representation is inadequate for a wide class of problems, e.g., knotted vortex filaments and points of vanishing vorticity [21,26,27,30,56]. The issue is also closely related to the non-uniqueness of potentials representing a given *u* (gauge transformations), which has been well known for many years [18] but has only recently been analyzed in depth [27–29]. A more detailed discussion of this matter is out of the scope of the present work. It is our plan to come back to this issue in the near future.

### 4.4. On the Representation of the Fluid Pressure by Means of Potentials

Apart from the representation of the velocity field, Equation (32), an explicit representation for the pressure field, in terms of the potentials $k$, $A_i$, and $a_i$, can be easily found out from Equation (31). Indeed, using Equation (5), we see that $\rho\,\partial E/\partial \rho = p/\rho$. Combining the latter with Equation (31), we obtain:

$$\frac{p}{\rho} = \frac{D\,k}{D\,t} - A\,\frac{D\,a}{D\,t} + \frac{u^2}{2} - E - P. \tag{33a}$$

An alternative representation of the pressure occurs again using the expression $\rho\,\partial E/\partial \rho = p/\rho$, in conjunction with Equation (32). Then, after straightforward calculations, we obtain:

$$\frac{p}{\rho} = \frac{\partial\,k}{\partial\,t} - A\,\frac{\partial\,a}{\partial\,t} - \frac{u^2}{2} - E - P. \tag{33b}$$

The right-hand side of the latter equation, where $u = u\,(k,\,A_i,\,a_i)$, is closely related (although more general) to the Lagrangian density provided in works relevant to the approach of Clebsch [37]; see, also, [57]. Further, Equations (33) are essential for the establishment of the connection between Equation (7) and the two alternative forms of the dynamic free-surface condition that occur variationally in Section 5; see Equations (51) and (52).

### 5. Boundary-Variational Equation: Derivation of Boundary Conditions

Having obtained the Euler–Lagrange equations within the fluid domain, that is, those equations corresponding to independent variations of the fields $k$, $A$, $a$, $\rho$, and *u* in $V\,(t)$, there remains the treatment of the boundary terms of the variational equation and deduction the boundary conditions. After substituting Equations (28)–(32) into the total variational Equation (27), the volume-integral terms are eliminated, and what remains is the following *boundary-variational equation*:

$$\int_T \int_D \left\{ \left[ \left( \frac{\partial\,\eta}{\partial\,t} - u\,N_\eta \right) (k\,\delta\rho + \rho\,A\,\delta a) - \rho\,k\,\delta u\,N_\eta \right]_{z\,=\,\eta} \right.$$

$$+ \left. \left[ \rho \left( \frac{u^2}{2} - E - P \right) - k \left( \frac{\partial\,\rho}{\partial\,t} + \nabla \cdot (\rho\,u) \right) - \rho\,A\,\frac{D\,a}{d\,t} - \overline{p} \right]_{z\,=\,\eta} \delta\eta \right\} dx\,dt$$

$$+ \int_T \int_D \left\{ \left( \frac{\partial\,h}{\partial\,t} - u\,N_h \right) (k\,\delta\rho + \rho\,A\,\delta a) - \rho\,k\,\delta u\,N_h \right\}_{z\,=\,-h} dx\,dt$$

$$- \int_T \int_{\partial D_w} \int_{-h\,(x,\,t)}^{\eta\,(x,\,t)} \left\{ (k\,\delta\rho + \rho\,A\,\delta a)\,u\,n_w + \rho\,k\,\delta u\,n_w \right\} dz\,dl\,dt = 0.$$

Using standard variational arguments, the above equation can be equivalently written as three independent variational equations, associated with the free surface, seabed, and lateral boundary, respectively:

$$
\int_T \int_D \left\{ \left[ \left( \frac{\partial \eta}{\partial t} - \boldsymbol{u}\, \boldsymbol{N}_\eta \right) (k\, \delta\rho + \rho\, A\, \delta a) - \rho\, k\, \delta\boldsymbol{u}\, \boldsymbol{N}_\eta \right]_{z\,=\,\eta} \right.
$$
$$
\left. + \left[ \rho \left( \frac{u^2}{2} - E - P \right) - k \left( \frac{\partial \rho}{\partial t} + \nabla \cdot (\rho\, \boldsymbol{u}) \right) - \rho\, A\, \frac{D\, a}{d\, t} - \overline{p} \right]_{z\,=\,\eta} \delta\eta \right\} dx\, dt = 0,
$$
(34a)

$$
\int_T \int_D \left\{ \left( \frac{\partial h}{\partial t} - \boldsymbol{u}\, \boldsymbol{N}_h \right) (k\, \delta\rho + \rho\, A\, \delta a) - \rho\, k\, \delta\boldsymbol{u}\, \boldsymbol{N}_h \right\}_{z\,=\,-h} dx\, dt = 0
$$
(34b)

and

$$
\int_T \int_{\partial D_w} \int_{-h(x,t)}^{\eta(x,t)} \left\{ (k\, \delta\rho + \rho A\, \delta a)\, \boldsymbol{u}\, \boldsymbol{n}_w + \rho\, k\, \delta\boldsymbol{u}\, \boldsymbol{n}_w \right\} dz\, dl\, dt = 0.
$$
(34c)

According to the standard practice in Hamiltonian variational principles, Equations (34a)–(34c) are expected to generate appropriate boundary conditions for the studied problem. Nevertheless, something curious happens in the present case. To clarify the situation, it is sufficient to restrict our attention to the free surface, Equation (34a). If one assumes that the Eulerian variations $\delta\rho$, $\delta a$, $\delta\boldsymbol{u}$, and $\delta\eta$ *are* independent of each other, then each one of the variations $\delta\rho$, $\delta a_i$, $i = 1,\ 2,\ 3$, considered separately, leads to the (same) kinematic free-surface condition:

$$
\frac{\partial \eta}{\partial t} - \boldsymbol{u}\, \boldsymbol{N}_\eta = 0.
$$

Although the above condition is correct (and expected), it is derived repetitively, which implies a redundancy of the assumed "independent" variations. The main issue, however, lies in the consideration of independent $\delta\boldsymbol{u}$ and $\delta\eta$. If the above kinematic condition is substituted into Equation (34a), and the mass and identity conservations, Equations (3) and (11), are also used, then there remains:

$$
\int_T \int_D \left\{ -\rho\, k\, \delta\boldsymbol{u}\, \boldsymbol{N}_\eta + \left[ \rho \left( \frac{u^2}{2} - E - P \right) - \overline{p} \right] \delta\eta \right\}_{z\,=\,\eta} dx\, dt = 0.
$$

From the above variational equation, with $\delta\boldsymbol{u}$ and $\delta\eta$ independent, it does not seem possible to derive the correct dynamic free-surface condition, even though some of its "ingredients" are present in the coefficients.

These observations suggest that the variations $\delta a$, $\delta\boldsymbol{u}$, $\delta\rho$, $\delta\eta$ on the boundary $\partial V(t)$ of the fluid cannot be considered independent. This is the motivation behind the introduction of the differential-variational relations, Equations (19) and (20). The latter provide rational connections between the Eulerian variations, which turn out to be sufficient for the correct and consistent derivation of all the boundary conditions.

*5.1. Transformation of the Boundary-Variational Equation Using the differential-Variational Constraints*

The use of the differential-variational relations (19)–(20) in the boundary-variational Equations (34a)–(34c) results in re-expressing the variations of the Eulerian fields in terms of the virtual displacements $\delta_L \boldsymbol{X}_b$, $b \in \{\eta, h, w\}$, on each boundary. Though, some attention is required for the substitution of $\delta\rho$, $\delta a$, and $\delta\boldsymbol{u}$ in Equations (34a) and (34b), referring to the free surface and the seabed. These field variations are generally functions of the variables $x$, $z$, and $t$, where, in this instance, the vertical variable is evaluated on $z = \eta$ or $z = -h$. On the other hand, the virtual displacements $\delta_L \boldsymbol{X}_b$, $b \in \{\eta, h\}$ are—by their nature—position variations of the material parcels that form the free-surface and seabed

surfaces and, thus, they are independent of $z$. Consequently, the differential-variational constraints (19a) and (19b) on the free surface and the seabed should be reformulated as:

$$
\begin{aligned}
[\delta\rho]_{z=\{\eta,-h\}} &= -[\nabla\rho]_{z=\{\eta,-h\}} \cdot \delta_L X_{\{\eta,h\}} - [\rho]_{z=\{\eta,-h\}} (\nabla \cdot \delta_L X_{\{\eta,h\}}) = \\
&= -[\nabla\rho]_{z=\{\eta,-h\}} \cdot \delta_L X_{\{\eta,h\}} - [\rho]_{z=\{\eta,-h\}} \left(\nabla_2 \cdot (\delta_L X_{\{\eta,h\},1}, \delta_L X_{\{\eta,h\},2})\right),
\end{aligned}
\tag{35}
$$

$$
[\delta a]_{z=\{\eta,-h\}} = -[\nabla a]_{z=\{\eta,-h\}} \cdot \delta_L X_{\{\eta,h\}},
\tag{36}
$$

where $\nabla_2(\cdot) \equiv (\partial/\partial x_1, \partial/\partial x_2)(\cdot)$ is the 2D-horizontal gradient. Further, using Equation (19c), the product $\delta u N_b, b \in \{\eta, h\}$ is written as:

$$
[\delta u]_{z=-h} N_h = -\left(\frac{\partial N_h}{\partial t} + [\nabla(u N_h)]_{z=-h}\right) \delta_L X_h, \quad z = -h,
\tag{37}
$$

recalling Equation (21), and:

$$
\begin{aligned}
[\delta u]_{z=\eta} N_\eta &= \frac{\partial(\delta_L X_\eta N_\eta)}{\partial t} + [(u_1, u_2)]_{z=\eta} \cdot \nabla_2(\delta_L X_\eta N_\eta) \\
&\quad - \left(\frac{\partial N_\eta}{\partial t} + [\nabla(u N_\eta)]_{z=\eta}\right) \delta_L X_\eta, \quad z = \eta.
\end{aligned}
\tag{38}
$$

For the same product on the lateral rigid-wall boundary, we have that:

$$
\delta u n_w = -\nabla(u n_w) \delta_L X_w, \quad (x, z) \in \partial V_w,
\tag{39}
$$

as is readily found using Equation (22) and the fact that $n_w$ is independent of time ($\partial V_w$ is a known fixed boundary).

Now, substituting the variations $\delta\eta$, $\delta\rho$, $\delta a$, and $\delta u$, in Equation (34a) with Equations (20), (35), (36), and (38), respectively, we obtain:

$$
\begin{aligned}
\int_T \int_D &\left\{ \left[ -\left(\frac{\partial\eta}{\partial t} - u N_\eta\right)(k\nabla\rho + \rho A\nabla a) + \rho k\left(\frac{\partial N_\eta}{\partial t} + \nabla(u N_\eta)\right) \right]_{z=\eta} \delta_L X_\eta \right. \\
&+ \left[ \rho\left(\frac{u^2}{2} - E - P\right) - k\left(\frac{\partial\rho}{\partial t} + \nabla\cdot(\rho u)\right) - \rho A \frac{Da}{dt} - \bar{p} \right]_{z=\eta} \delta_L X_\eta N_\eta \\
&- \left[ \rho k\left(\frac{\partial\eta}{\partial t} - u N_\eta\right) \right]_{z=\eta} \nabla_2 \cdot (\delta_L X_{\eta,1}, \delta_L X_{\eta,2}) \\
&\left. - [\rho k]_{z=\eta} \frac{\partial}{\partial t}(\delta_L X_\eta N_\eta) - [\rho k(u_1, u_2)]_{z=\eta} \nabla_2(\delta_L X_\eta N_\eta) \right\} dx\, dt = 0
\end{aligned}
\tag{40}
$$

on the free surface, in terms of the virtual displacements $\delta_L X_\eta$. Similarly, using the differential-variational relations on the seabed and the lateral rigid wall, Equations (34b) and (34c) are, respectively, rewritten as:

$$
\begin{aligned}
\int_T \int_D &\left\{ \left[ -\left(\frac{\partial h}{\partial t} - u N_h\right)(k\nabla\rho + \rho A\nabla a) + \rho k\left(\frac{\partial N_h}{\partial t} + \nabla(u N_h)\right) \right]_{z=-h} \delta_L X_h \right. \\
&\left. - \left[ \rho k\left(\frac{\partial h}{\partial t} - u N_h\right) \right]_{z=-h} \nabla_2 \cdot (\delta_L X_{h,1}, \delta_L X_{h,2}) \right\} dx\, dt = 0,
\end{aligned}
\tag{41}
$$

involving the variations $\delta_L X_h$ of the seabed parcels, and:

$$
\begin{aligned}
\int_T \int_{\partial D_w} \int_{-h(x,t)}^{\eta(x,t)} &\left\{ -[(u n_w)(k\nabla\rho + \rho A\nabla a) + \rho k\nabla(u n_w)]\, \delta_L X_w \right. \\
&\left. - \rho k(u n_w)\nabla\cdot(\delta_L X_w) \right\} dz\, dl\, dt = 0,
\end{aligned}
\tag{42}
$$

involving the variations $\delta_L X_w$ of the parcels on the rigid wall.

*5.2. Decomposition of Boundary Virtual Displacements into Normal and Tangential Components*

To facilitate the further treatment of the variational Equations (40)–(42), we analyze each $\delta_L X_b$, $b \in \{ \eta, h, w \}$ into its normal and tangential components, i.e., we write:

$$\delta_L X_b = \delta_L X_{b, \perp} + \delta_L X_{b, \parallel}, \ b \in \{ \eta, h, w \}. \tag{43}$$

The normal components $\delta_L X_{b, \perp}$ are expressed as:

$$\begin{cases} \delta_L X_{b, \perp} = \delta B_{b, \perp} \dfrac{N_b}{\|N_b\|^2}, \ b \in \{ \eta, h \}, \\ \delta_L X_{w, \perp} = \delta B_{w, \perp} \, \boldsymbol{n}_w , \end{cases} \tag{44}$$

where $\delta B_{b, \perp}$ are scalar quantities (On the free surface and the seabed, it is convenient to express the normal direction with the vector $N_b \, / \, \|N_b\|^2$ instead of the unit vector $\boldsymbol{n}_b = N_b \, / \, \|N_b\|$. This is carried out to simplify some expressions in the subsequent algebraic manipulations. Its effect is an indifferent rescaling of the scalar $\delta B_{b, \perp}$).

The tangential component $\delta_L X_{b, \parallel}$ is expanded in the natural basis of the tangent planes. Considering the parametric representations of the free surface and the seabed, at frozen time $t = \tilde{t}$:

$$\boldsymbol{r}_\eta (\boldsymbol{x}) = ( x_1, x_2, \eta (x_1, x_2, \tilde{t}) ), \ \boldsymbol{r}_h (\boldsymbol{x}) = ( x_1, x_2, - h (x_1, x_2, \tilde{t}) ), \tag{45a}$$

we obtain the two tangent vectors:

$$\boldsymbol{T}_{b, 1} \equiv \frac{\partial \, \boldsymbol{r}_b}{\partial \, x_1}, \ \boldsymbol{T}_{b, 2} \equiv \frac{\partial \, \boldsymbol{r}_b}{\partial \, x_2}, \ b \in \{ \eta, h \}, \tag{45b}$$

which constitute a natural local basis of the tangent plane at each point of the boundary surface. As concerns the lateral rigid-wall boundary, being a fixed vertical surface, it has a unit normal vector $\boldsymbol{n}_w (\boldsymbol{x}) = (n_{w, 1}, n_{w, 2}, 0) (\boldsymbol{x})$ and respective unit tangent vectors:

$$\boldsymbol{T}_{w, 1} \equiv (0, 0, 1), \ \boldsymbol{T}_{w, 2} \equiv (n_{w, 2}, - n_{w, 1}, 0). \tag{45c}$$

Therefore, the tangent component of the virtual displacements, $\delta_L X_{b, \parallel}$, may be generally written as:

$$\delta_L X_{b, \parallel} = \delta B_{b, 1} \, \boldsymbol{T}_{b, 1} + \delta B_{b, 2} \, \boldsymbol{T}_{b, 2}, \ b \in \{ \eta, h, w \}, \tag{46}$$

where $\delta B_{b, 1}$ and $\delta B_{b, 2}$ are appropriate scalar quantities.

Adopting this decomposition for $\delta_L X_b$ leads to independent variations of the scalars $\delta B_{b, \{ \perp, 1, 2 \}}$, in place of the component variations $(\delta_L X_{b, 1}, \delta_L X_{b, 2}, \delta_L Z_b)$.

*5.3. Free-Surface Conditions*

Introducing the representations (43) and (46) into the free-surface variational Equation (40), and considering, first, tangent variations to the boundary ($\delta_L X_{\eta, \parallel}$ = arbitrary, $\delta_L X_{\eta, \perp} = 0$), we obtain (note that $\delta_L X_{\eta, \parallel} \cdot N_\eta = 0$):

$$\int_T \int_D \left[ \rho k \left( \frac{\partial N_\eta}{\partial t} + \nabla (\boldsymbol{u} \, N_\eta) \right) - \left( \frac{\partial \eta}{\partial t} - \boldsymbol{u} \, N_\eta \right) (k \, \nabla \rho + \rho \, A \, \nabla a) \right]_{z \, = \, \eta} \delta_L X_{\eta, \parallel} \, d\boldsymbol{x} \, dt$$

$$- \int_T \underbrace{\int_D \left[ \rho \, k \left( \frac{\partial \eta}{\partial t} - \boldsymbol{u} \, N_\eta \right) \right]_{z \, = \, \eta} \nabla_2 \cdot (\delta B_{\eta, 1}, \delta B_{\eta, 2}) \, d\boldsymbol{x}}_{I_\eta} \, dt = 0 \tag{47}$$

Invoking the 2D divergence theorem for the integral $I_\eta$ of Equation (47), and neglecting subsequent terms on the line boundary of the free surface (boundary of co-dimension 2), we find:

$$I_\eta \ = \ - \int_D \nabla_2 \left( \left[ \rho \, k \left( \frac{\partial \eta}{\partial t} \ - \ \boldsymbol{u} \, \boldsymbol{N}_\eta \right) \right]_{z \, = \, \eta} \right) \ (\delta \, B_{\eta, \ 1}, \ \delta \, B_{\eta, \ 2}) \ d \, \boldsymbol{x}.$$

Notice that the above integrand involves a term where the evaluation on $z = \eta$ precedes the differentiation with $\nabla_2 \, (\cdot)$. This originates from the use of Equations (35), (36), and (38) for the *a priori* evaluated field variations on the boundary, and it is in contrast to the situation observed in the first integral of Equation (47), where similar terms are first differentiated and afterwards evaluated on the free surface. The following lemma will facilitate the calculation of terms $\nabla_2 \, (\cdot)$, appearing in the integrand of $I_\eta$ and in the integrand of a similar integral associated with the seabed, below.

**Lemma 1.** For any sufficiently smooth function $f \ = \ f \, (\boldsymbol{x}, \, z, \, t)$ of the flow, it holds that:

$$\nabla_2 \left( [ f \, ]_{z \, = \, \{ \, \eta, \, - \, h \, \}} \right) (\delta \, B_{\{ \, \eta, \, h \, \}, \, 1}, \ \delta \, B_{\{ \, \eta, \, h \, \}, \, 2}) \ =$$
$$= \ [\nabla f]_{z = \{\eta, -h\}} \ (\delta \, B_{\{ \, \eta, \, h \, \}, \, 1} \ \boldsymbol{T}_{\{ \, \eta, \, h \, \}, \, 1} \ + \ \delta \, B_{\{ \, \eta, \, h \, \}, \, 2} \ \boldsymbol{T}_{\{ \, \eta, \, h \, \}, \, 2}) \, .$$

The proof of the lemma is given in Appendix B.

Given the above Lemma 1, $I_\eta$ becomes:

$$I_\eta \ = \ - \int_D \left\{ \nabla \left[ \rho \, k \left( \frac{\partial \eta}{\partial t} \ - \ \boldsymbol{u} \, \boldsymbol{N}_\eta \right) \right] \right\}_{z \, = \, \eta} (\delta \, B_{\eta, \ 1} \ \boldsymbol{T}_{\eta, \, 1} \ + \ \delta \, B_{\eta, \ 2} \ \boldsymbol{T}_{\eta, \, 2}) \ d \, \boldsymbol{x}.$$

Substituting the latter into Equation (47), and using the following relation (recall Equation (2a) for $\boldsymbol{N}_\eta$):

$$\frac{\partial \, \boldsymbol{N}_\eta}{\partial t} \ + \ \nabla \, (\boldsymbol{u} \, \boldsymbol{N}_\eta) \ + \ \nabla \left( \frac{\partial \eta}{\partial t} \ - \ \boldsymbol{u} \, \boldsymbol{N}_\eta \right) \ = \ \frac{\partial \, \boldsymbol{N}_\eta}{\partial t} \ + \ \nabla \left( \frac{\partial \eta}{\partial t} \right) \ = \ 0,$$

we obtain the final form of the free-surface variational equation for tangential virtual displacements, reading as:

$$\int_T \int_D \left[ \rho \left( \frac{\partial \eta}{\partial t} \ - \ \boldsymbol{u} \, \boldsymbol{N}_\eta \right) (- \ \nabla \, k \ + \ \boldsymbol{A} \, \nabla \, a) \right]_{z \, = \, \eta} (\delta \, B_{\eta, \ 1} \ \boldsymbol{T}_{\eta, \, 1} \ + \ \delta \, B_{\eta, \ 2} \ \boldsymbol{T}_{\eta, \, 2}) \ d \, \boldsymbol{x} \, d \, t \ = \ 0.$$

Therefore, since $\rho \, \neq \, 0, \boldsymbol{u} \ = \ - \, \nabla \, k \ + \ \boldsymbol{A} \, \nabla \, a$, and the variations $\delta \, B_{\eta, \, 1}$ and $\delta \, B_{\eta, \, 2}$ are independent and arbitrary, we obtain the Euler–Lagrange equations:

$$\left( \frac{\partial \eta}{\partial t} \ - \ \boldsymbol{u} \, \boldsymbol{N}_\eta \right) \boldsymbol{u} \, \boldsymbol{T}_{\eta, \, i} \ = \ 0, \, i \ = \ 1, \, 2, \, z \ = \ \eta \, (\boldsymbol{x}, \, t). \tag{48}$$

Equation (48) provide us with two (non-exclusive) possibilities: either $\partial \, \eta \, / \, \partial \, t \ - \ \boldsymbol{u} \, \boldsymbol{N}_\eta \ = \ 0$ or $u_\parallel$ =magnitude of the tangential velocity $ = \ 0$. Since, on the free surface, the tangent velocity cannot generally be zero, we conclude that:

$$\delta_L \, \boldsymbol{X}_{\eta, \ \parallel} \ : \qquad \frac{\partial \eta}{\partial t} \ - \ \boldsymbol{u} \, \boldsymbol{N}_\eta \ = \ 0, \, z \ = \ \eta, \tag{49}$$

which constitutes the *free-surface kinematic condition*, coinciding with Equation (6).

Now, we return to the free-surface variational Equation (40), considering normal variations to the boundary ($\delta_L \, \boldsymbol{X}_{\eta, \ \perp} \ = $ arbitrary, $\delta_L \, \boldsymbol{X}_{\eta, \ \parallel} \ = \ 0$). Given Equation (44) and

the kinematic condition Equation (49), which also implies that $\partial N_\eta / \partial t + \nabla (u N_\eta) = 0$, Equation (40) becomes:

$$\int_T \int_D \left[ \rho \left( \frac{u^2}{2} - E - P \right) - k \left( \frac{\partial \rho}{\partial t} + \nabla \cdot (\rho u) \right) - \rho A \frac{da}{dt} - \overline{p} \right]_{z = \eta} \delta B_{\eta, \perp} \, dx \, dt$$

$$- \underbrace{\int_T \int_D [\rho k]_{z = \eta} \frac{\partial}{\partial t} (\delta B_{\eta, \perp}) \, dx \, dt}_{J_{\eta, 1}} - \underbrace{\int_T \int_D [\rho k (u_1, u_2)]_{z = \eta} \nabla_2 (\delta B_{\eta, \perp}) \, dx \, dt}_{J_{\eta, 2}} = 0$$

Using the isochronality condition for the integral $J_{\eta, 1}$, and the 2D divergence theorem for the integral $J_{\eta, 2}$, we obtain:

$$\int_T \int_D \left[ \rho \left( \frac{u^2}{2} - E - P \right) - k \left( \frac{\partial \rho}{\partial t} + \nabla \cdot (\rho u) \right) - \rho A \frac{da}{dt} - \overline{p} \right]_{z = \eta} \delta B_{\eta, \perp} \, dx \, dt$$

$$+ \int_T \int_D \left\{ \frac{\partial}{\partial t} \left( [\rho k]_{z = \eta} \right) + \nabla_2 \cdot \left( [\rho k (u_1, u_2)]_{z = \eta} \right) \right\} \delta B_{\eta, \perp} \, dx \, dt = 0 , \tag{50}$$

where, again, terms on the free surface's line boundary are neglected. The following lemma permits us to reformulate the integrand of the second integral in Equation (50) and simplify the calculations.

**Lemma 2.** Given the free-surface kinematic condition Equation (49), any sufficiently smooth function $f = f(x, z, t)$ of the flow satisfies the relation:

$$\frac{\partial}{\partial t} \left( [f]_{z = \eta} \right) + \nabla_2 \cdot \left( [f u_{2D}]_{z = \eta} \right) = \left[ \frac{\partial f}{\partial t} + \nabla \cdot (f u) \right]_{z = \eta},$$

where $u_{2D} = (u_1, u_2)$ is the horizontal fluid velocity. The proof of the lemma is given in Appendix B.

Applying Lemma 2 for $f = \rho k$, and after simple calculations, Equation (50) is finally shaped as:

$$\int_T \int_D \left\{ \rho \frac{Dk}{Dt} - \rho A \frac{Da}{Dt} + \rho \left( \frac{u^2}{2} - E - P \right) - \overline{p} \right\}_{z = \eta} \delta B_{\eta, \perp} \, dx \, dt = 0.$$

Thus, for arbitrary variations $\delta B_{\eta, \perp}$, we obtain the *free-surface dynamic condition*:

$$\delta_L X_{\eta, \perp} : \quad - \frac{Dk}{Dt} + A \frac{Da}{Dt} - \frac{u^2}{2} + E + P = - \frac{\overline{p}}{\rho}, \quad z = \eta. \tag{51}$$

Recalling the pressure representation Equation (33a), the above condition is rewritten in the form $p = \overline{p}$, coinciding with Equation (7).

**Remark 5.** If we take into account the velocity representation (32), then the free-surface dynamic condition, Equation (51), is written as:

$$- \frac{\partial k}{\partial t} + A \frac{\partial a}{\partial t} + \frac{u^2(k, A_i, a_i)}{2} + E + P = - \frac{\overline{p}}{\rho}, \quad z = \eta, \tag{52}$$

which, recalling the pressure representation (33b), becomes again $p = \overline{p}$. The left-hand side of Equation (52) is closely related to the Lagrangian density proposed by Clebsch [37] for the variational formulation of Euler equations using Clebsch potentials; see, also, [57]. What differs, obviously, is the presence of additional terms, which is due to the additional features of compressibility, conservative body forces, and applied pressure on the free surface.

**Remark 6.** The free-surface part of the problem is also dealt with by Timokha [40] and Berdichevsky [31] (Section 9.3) but for incompressible fluids. (a) Timokha uses the unconstrained Clebsch–Bateman–Luke principle (not Hamilton's principle, as herein) for the case of wave sloshing, where the Lagrangian density is a modification of Equation (52) involving a single label (original Clebsch representation). (b) Berdichevsky begins with Hamilton's principle but *a priori imposes the free-surface kinematic condition* as essential. Additionally, he does not explicitly introduce the free-surface elevation $\eta$. To arrive at the dynamic condition on the free surface, he identifies the connection between the variations of the velocity and of the free-surface boundary via Equation (19c). His result is similar to Equation (51) but without the term $A \, D \, a \, / \, d \, t$, which is added artificially (on the basis of the identity conservation) in order to derive Equation (52).

*5.4. Seabed and Lateral Rigid-Wall Conditions*

In the seabed variational Equation (41), Equation (43) reduces to $\delta_L \, X_h \; = \; \delta_L \, X_{h, \, \parallel}$ because of Equation (21). Specifically, on the seabed, we have to consider only tangential variations for which Equation (41) reads:

$$\int_T \int_D \Big\{ \Big[ \rho \, k \, \Big( \frac{\partial \, \mathbf{N}_h}{\partial \, t} \; + \; \nabla \, (u \; \mathbf{N}_h) \Big) \; - \; \Big( \frac{\partial \, h}{\partial \, t} \; - \; u \, \mathbf{N}_h \Big) \, (k \, \nabla \rho \; + \; \rho \, A \, \nabla \, a) \Big]_{z \, = \, - \, h} \, \delta_L \, x_{h, \, \parallel}$$
$$- \, \Big[ \rho \; k \, \Big( \frac{\partial \, h}{\partial \, t} \; - \; u \; \mathbf{N}_h \Big) \Big]_{z \, = \, - \, h} \nabla_2 \; \cdot \; (\delta \, B_{h, \, 1}, \, \delta \, B_{h, \, 2}) \Big\} d \, x \; d \, t \; = \; 0 ,$$

with $\delta_L \, X_{h, \, \parallel} \; = \; \delta \, B_{h, \, 1} \, T_{h, \, 1} \; + \; \delta \, B_{h, \, 2} \, T_{h, \, 2}$; see Equations (45a), (45b) and (46). This variational equation is of exactly the same structure as Equation (47) above, having the index $h$ in the place of $\eta$, and with evaluation of the terms on $z \; = \; - \, h$ in the place of $z \; = \; \eta$. Thus, we treat it in a similar way, concluding to the final form:

$$\int_T \int_D \Big[ \rho \, \Big( \frac{\partial \, h}{\partial \, t} \; - \; u \, \mathbf{N}_h \Big) \, (- \, \nabla \, k \; + \; A \, \nabla \, a) \Big]_{z \; = \; - \, h} \, (\delta \, B_{h, \, 1} \, T_{h, \, 1} \; + \; \delta \, B_{h, \, 2} \, T_{h, \, 2}) \; d \, x \; d \, t \; = \; 0.$$

As before, since $u \; = \; - \, \nabla \, k \; + \; A \, \nabla \, a$, and the variations $\delta \, B_{h, \, 1}$ and $\delta \, B_{h, \, 2}$ are independent and arbitrary, we obtain the Euler–Lagrange equations:

$$\Big( \frac{\partial \, h}{\partial \, t} \; - \; u \; \mathbf{N}_h \Big) \, u \; T_{h, \, i} \; = \; 0, \, i \; = \; 1, \, 2, \, z \; = \; - \, h \, (x, \, t). \tag{53}$$

Thus, thinking as in the case of the free surface, we conclude to the *moving-seabed kinematic condition*:

$$\delta_L \, X_{h, \, \parallel} \; : \; \frac{\partial \, h}{\partial \, t} \; - \; u \, \mathbf{N}_h \; = \; 0, \, z \; = \; - \, h. \tag{54}$$

Obviously, this condition coincides with Equation (8).

Last, on the lateral rigid wall, owing to Equation (22), the variations are again only tangential, that is, $\delta_L \, X_w \; = \; \delta_L \, X_{w, \, \parallel}$, given by Equation (45c)–(46). Accordingly, after simple algebraic calculations, the variational Equation (42) becomes:

$$- \int_T \int_{\partial \, D_w} \int_{- \, h \, (x, \, t)}^{\eta \, (x, \, t)} \rho \, (u \; \mathbf{n}_w) \, (- \, \nabla \, k \; + \; A \, \nabla \, a) \, \delta_L \, x_{w, \, \parallel} \; d \, z \; d \, l \; d \, t$$

$$- \int_T \int_{\partial \, D_w} \int_{- \, h \, (x, \, t)}^{\eta \, (x, \, t)} \nabla \; \cdot \; \Big\{ \rho \, k \, (u \; \mathbf{n}_w) \, \delta_L \, x_{w, \, \parallel} \Big\} \; d \, z \; d \, l \; d \, t \; = \; 0$$

The second integral, though, is just the divergence of a vector field over the surface $\partial \, V_w$. Thus, it is omitted, integrating out to the line boundary of the lateral rigid wall. As a result, we obtain the following Euler–Lagrange equations:

$$(u \; \mathbf{n}_w) \, u \; T_{w, \, i} \; = \; 0, \, i \; = \; 1, \, 2, \, (x, \, z) \; \in \; \partial \, V_w, \tag{55}$$

which lead to the *impermeability condition* (9):

$$\delta_L \, \boldsymbol{X}_{\text{w}, \, \parallel} \; : \; \boldsymbol{u} \, \boldsymbol{n}_{\text{w}} \; = \; 0, \; (\boldsymbol{x}, z) \; \in \; \partial V_{\text{w}}, \tag{56}$$

using the same reasoning as previously. With the latter condition, we complete the variational derivation of all the boundary conditions that are involved in the differential formulation of the problem (Section 2.3) using Hamilton's principle.

**Remark 7.** Interestingly enough, we observe that the variational derivation of kinematic boundary conditions reveals a new possibility: the tangential velocity on the boundary may either be as in irrotational flow (no condition) or may be zero; see Equations (48), (53), and (55). Without being able to discuss this new possibility in depth, we mention that the zero tangential velocity is compatible with the zero-viscosity limit of the Navier–Stokes equations.

## 6. Discussion and Conclusions

In this work, the Herivel–Lin variational approach for rotational flows, based on Hamilton's principle, was extended to the case of rotational water waves over a moving seabed. The original contribution of the present work is the variational derivation of the complete set of governing equations and boundary conditions, including both kinematic and dynamic free-surface conditions. A possible explanation for the lack of such a derivation in the relevant literature might be the inadequacy of integral constraints to resolve the variational controversy on the boundary, although they do so for the bulk fluid motion.

To address this issue, the constraints of mass and parcels' identity conservation are reconsidered in a stronger (pointwise) version. The latter takes the form of differential-variational relations between the variations of the Eulerian fields and the virtual displacements of fluid parcels. Substituting these relations on the boundary leads to a reformulation of the boundary variational equation, which provides us with the correct boundary conditions.

This somewhat unconventional, variational procedure succeeds in two main points. First, besides the derivation of the full equations for the bulk fluid motion, convenient representations of the velocity and pressure fields are obtained in terms of Clebsch–Lin potentials. This is possible because of the ability to consider independent variations of the Eulerian fields in the interior of the fluid domain, owing to the inclusion of the conservation of mass and the conservation of parcels' identity as integral constraints applied to the action functional. Second, re-expressing the boundary variational equation in terms of the fluid parcels' virtual displacements leads to a successful derivation of both the kinematic and dynamic boundary conditions. To our knowledge, this has never been carried out before for such flows using Hamilton's principle.

The present variational formulation, however complicated might be, offers valuable information concerning the variational treatment of rotational water waves. Though, certain concepts that are involved require further clarification. For example, (i) the two-level constraints (integral in the fluid volume, differential on the fluid boundary), introduced herein for the first time, call for a better understanding and theoretical assessment; (ii) the gauge freedom of the Clebsch–Lin potentials and the ability of the latter to represent generic rotational flows need to be clarified. It is the authors' intention to address those issues in the near future. Nevertheless, we believe that the outcome of this work is a decisive step towards a subsequent formulation of an unconstrained variational principle, which will incorporate all the essential features of rotational free-surface flows.

**Author Contributions:** Conceptualization, methodology, formal analysis, investigation, writing—original draft preparation, writing—review and editing: C.P.M. and G.A.A. contributed equally to this work, being in continuous collaboration. All authors have read and agreed to the published version of the manuscript.

**Funding:** This research received no external funding.

**Institutional Review Board Statement:** Not applicable.

**Informed Consent Statement:** Not applicable.

**Data Availability Statement:** Not applicable.

**Acknowledgments:** C. P. Mavroeidis is supported by the ELKE-NTUA scholarship program.

**Conflicts of Interest:** The authors declare no conflict of interest.

## Appendix A. Detailed Calculation of the Action Functional's Partial Gateaux Derivatives

In this Appendix, we derive the partial Gateaux derivatives (26c), (26d), and (26e), of the action functional (16), Section 4.1, with respect to the fields $a\,(x,\,z,\,t)$, $\rho\,(x,\,z,\,t)$, and $u\,(x,\,z,\,t)$.

***Variation with respect to the parcel labels***: Considering the variation $\delta_a\,(\cdot)$ of the action functional (16), we have that:

$$
\begin{aligned}
\delta_a\,\widetilde{\mathscr{S}} \;=\; &-\int_T\int_D\int_{-\,h\,(x,\,t)}^{\eta\,(x,\,t)}\;\rho\,A\,\frac{D\,(\delta\,a)}{D\,t}\;dz\;dx\;dt \;\;= \\[2mm]
\;=\;&\int_T\int_D\int_{-\,h\,(x,\,t)}^{\eta\,(x,\,t)}\;\frac{D\,(\rho\,A)}{D\,t}\,\delta\,a\;dz\;dx\;dt \\[2mm]
&-\underbrace{\int_T\int_D\int_{-\,h\,(x,\,t)}^{\eta\,(x,\,t)}\;\frac{D}{D\,t}\,(\rho\,A\,\delta\,a)\;dz\;dx\;dt}_{I}.
\end{aligned}
\tag{A1}
$$

The second integral of the rightmost-hand side above equation, $I$, can be further analyzed as:

$$
\begin{aligned}
I \;=\; &\int_T\int_D\int_{-\,h\,(x,\,t)}^{\eta\,(x,\,t)}\;\left(\tfrac{\partial}{\partial\,t}\,(\rho\,A\,\delta\,a)\;+\;u\,\nabla\,(\rho\,A\,\delta\,a)\right)\;dz\;dx\;dt \;\;= \\[2mm]
\;=\;&\underbrace{\int_T\int_D\int_{-\,h\,(x,\,t)}^{\eta\,(x,\,t)}\;\tfrac{\partial}{\partial\,t}\,(\rho\,A\,\delta\,a)\;dz\;dx\;dt}_{I_1}\;+\;\underbrace{\int_T\int_D\int_{-\,h\,(x,\,t)}^{\eta\,(x,\,t)}\;u\,\nabla\,(\rho\,A\,\delta\,a)\;dz\;dx\;dt}_{I_2}.
\end{aligned}
$$

Then, utilizing the Leibnitz integral rule for differentiation under the integral sign, and using the relation $u\,\nabla\,(\rho\,A\,\delta\,a)\;=\;\nabla\cdot((\rho\,A\,\delta\,a)\,u)\;-\;(\nabla\cdot u)\,\rho\,A\,\delta\,a$ for $I_2$, we obtain:

$$
I_1\;=\;\frac{\partial}{\partial\,t}\int_{-\,h\,(x,\,t)}^{\eta\,(x,\,t)}\rho\,A\,\delta\,a\;dz\;-\;[\,\rho\,A\,\delta\,a\,]_{z\,=\,\eta}\,\frac{\partial\,\eta}{\partial\,t}\;-\;[\,\rho\,A\,\delta\,a\,]_{z\,=\,-\,h}\,\frac{\partial\,h}{\partial\,t}
$$

and:

$$
\begin{aligned}
I_2 \;=\; &\int_{-h\,(x,\,t)}^{\eta\,(x,\,t)} \nabla \cdot [\,(\rho\,A\,\delta\,a)\,u\,]\,dz \;-\; \int_{-h\,(x,\,t)}^{\eta\,(x,\,t)} (\nabla \cdot u)\,\rho\,A\,\delta\,a\,dz \;=\; \\[2mm]
=\; &\sum_{i=1}^{2} \int_{-h\,(x,\,t)}^{\eta\,(x,\,t)} \frac{\partial}{\partial x_i}\,[\,(\rho\,A\,\delta\,a)\,u_i\,]\,dz \;+\; \int_{-h\,(x,\,t)}^{\eta\,(x,\,t)} \frac{\partial}{\partial z}\,[\,(\rho\,A\,\delta\,a)\,u_3\,]\,dz \;-\; \int_{-h\,(x,\,t)}^{\eta\,(x,\,t)} (\nabla \cdot u)\,\rho\,A\,\delta\,a\,dz \;=\; \\[2mm]
=\; &\sum_{i=1}^{2}\left( \frac{\partial}{\partial x_i} \int_{-h\,(x,\,t)}^{\eta\,(x,\,t)} (\rho\,A\,\delta\,a)\,u_i\,dz \right. \\[2mm]
&-\; [(\rho\,A\,\delta\,a)\,u_i]_{z\,=\,\eta}\,\frac{\partial \eta}{\partial x_i} \;-\; \left.[(\rho\,A\,\delta\,a)\,u_i]_{z\,=\,-h}\,\frac{\partial h}{\partial x_i}\right) \\[2mm]
&+\; [(\rho\,A\,\delta\,a)\,u_3]_{z\,=\,\eta} \;-\; [(\rho\,A\,\delta\,a)\,u_3]_{z\,=\,-h} \;-\; \int_{-h\,(x,\,t)}^{\eta\,(x,\,t)} (\nabla \cdot u)\,\rho\,A\,\delta\,a\,dz\,.
\end{aligned}
$$

Combining the above results, and rearranging various terms, the integral *I* finally becomes:

$$
\begin{aligned}
I \;=\; \int_{T}\int_{D}\Bigg\{ &\frac{\partial}{\partial t}\int_{-h\,(x,\,t)}^{\eta\,(x,\,t)} \rho\,A\,\delta\,a\,dz \;+\; \sum_{i=1}^{2} \frac{\partial}{\partial x_i}\int_{-h\,(x,\,t)}^{\eta\,(x,\,t)} (\rho\,A\,\delta\,a)\,u_i\,dz \\[2mm]
&-\; \int_{-h\,(x,\,t)}^{\eta\,(x,\,t)} (\nabla \cdot u)\,\rho\,A\,\delta\,a\,dz \;-\; \left(\frac{\partial \eta}{\partial t} - [u]_{z\,=\,\eta}\,N_\eta\right)[\rho\,A\,\delta\,a]_{z\,=\,\eta} \\[2mm]
&-\; \left(\frac{\partial h}{\partial t} - [u]_{z\,=\,-h}\,N_h\right)[\rho\,A\,\delta\,a]_{z\,=\,-h}\Bigg\}\,dx\,dt\,,
\end{aligned}
\tag{A2}
$$

recalling that $N_\eta$ and $N_h$ are outward normal vectors on the free surface and the seabed, respectively, given by Equations (2a) and (2b).

However, the first integral of the above equation integrates out to the boundaries of the time domain T and, thus, vanishes, according to the isochronality condition. Specifically:

$$
\int_{T}\int_{D} \frac{\partial}{\partial t}\int_{-h\,(x,\,t)}^{\eta\,(x,\,t)} \rho\,A\,\delta\,a\,dz\,dx\,dt \;=\; 0.
\tag{A3}
$$

As for the next term (the sum of the two integrals for *i* = 1,2), we may write:

$$
\begin{aligned}
I_{\partial V_{\mathrm{lat}}} \;\equiv\; &\int_{T}\int_{D}\sum_{i=1}^{2} \frac{\partial}{\partial x_i}\int_{-h\,(x,\,t)}^{\eta\,(x,\,t)} (\rho\,A\,\delta\,a)\,u_i\,dz\,dx\,dt \;=\; \\[2mm]
=\; &\int_{T}\int_{D} \nabla_2 \cdot G\,(x,\,t)\,dx\,dt\,,
\end{aligned}
\tag{A4}
$$

$\nabla_2\,(\cdot) \equiv (\,\partial\,(\cdot)\,/\,\partial x_1,\,\partial\,(\cdot)\,/\,\partial x_2\,)$ being the 2D gradient, where:

$$
G\,(x,\,t) \;=\; (G_1\,(x,\,t),\,G_2\,(x,\,t)),\; G_i\,(x,\,t) \;=\; \int_{-h\,(x,\,t)}^{\eta\,(x,\,t)} (\rho\,A\,\delta\,a)\,u_i\,dz.
$$

Then, invoking the divergence theorem in two dimensions, we obtain:

$$
I_{\partial V_{\mathrm{lat}}} \;=\; \int_{T}\int_{D} \nabla_2 \cdot G\,(x,\,t)\,dx\,dt \;=\; \int_{T}\oint_{\partial D} G\,(x,\,t)\,n_{\partial D}\,dl\,dt,
$$

where $n_{\partial D}\,(x) = (n_{\partial D,\,1},\,n_{\partial D,\,2})\,(x)$ is the outward unit normal vector on the boundary $\partial D$ of the horizontal domain. Introducing, next, the explicit form of $G\,(x,\,t)$ into the above, and rearranging terms, $I_{\partial V_{\mathrm{lat}}}$ is written as:

$$I_{\partial V_{\text{lat}}} = \int_T \oint_{\partial D} \left\{ \int_{-h(x,t)}^{\eta(x,t)} (\rho A \, \delta a) \, u_1 \, n_{\partial D,1} \, dz + \int_{-h(x,t)}^{\eta(x,t)} (\rho A \, \delta a) \, u_2 \, n_{\partial D,2} \, dz \right\} dl \, dt =$$

$$= \int_T \oint_{\partial D} \int_{-h(x,t)}^{\eta(x,t)} (\rho A \, \delta a) \, \{ u_1 \, n_{\partial D,1} + u_2 \, n_{\partial D,2} \} \, dz \, dl \, dt = \qquad (A5)$$

$$= \int_T \oint_{\partial D} \int_{-h(x,t)}^{\eta(x,t)} (\rho A \, \delta a) \, u \, n_{\text{lat}} \, dz \, dl \, dt,$$

where $n_{\text{lat}}(x) = (n_{\partial D,1}, n_{\partial D,2}, 0)(x)$ is the outward unit normal vector on the (vertical) lateral surface $\partial V_{\text{lat}}$.

Using, therefore, Equations (A2)–(A5) in Equation (A1), and given the fact that $\delta a$ vanish everywhere on $\partial V_{\text{lat}}$, except for the rigid-wall part $\partial V_w$ (see Sections 2.3 and 3.4), we finally obtain Equation (26c).

***Variation with respect to the fluid density***: For the variation $\delta_\rho(\cdot)$ of the action functional (16), we initially calculate:

$$\delta_\rho \widetilde{\mathscr{S}} = \int_T \int_D \int_{-h(x,t)}^{\eta(x,t)} \left\{ \left( \tfrac{1}{2} u^2 - E - P \right) \delta\rho - \rho \, \frac{\partial E}{\partial \rho} \, \delta\rho \right.$$

$$\left. - k \left( \frac{\partial(\delta\rho)}{\partial t} + u \cdot \nabla(\delta\rho) + \nabla \cdot u \, \delta\rho \right) - A \frac{D a}{D t} \delta\rho \right\} dz \, dx \, dt =$$

$$= \int_T \int_D \int_{-h(x,t)}^{\eta(x,t)} \left( \tfrac{1}{2} u^2 - E - P - \rho \frac{\partial E}{\partial \rho} - k \nabla \cdot u - A \frac{D a}{D t} \right) \delta\rho \, dz \, dx \, dt \qquad (A6)$$

$$- \underbrace{\int_T \int_D \int_{-h(x,t)}^{\eta(x,t)} k \, \frac{\partial(\delta\rho)}{\partial t} \, dz \, dx \, dt}_{J_1} - \underbrace{\int_T \int_d \int_{-h(x,t)}^{\eta(x,t)} k \, u \cdot \nabla(\delta\rho) \, dz \, dx \, dt}_{J_2}.$$

As concerns the above integrals $J_1$ and $J_2$, using the Leibnitz integral rule and the isochronality condition, we obtain:

$$J_1 = \int_{-h(x,t)}^{\eta(x,t)} \frac{\partial}{\partial t} (k \, \delta\rho) \, dz - \int_{-h(x,t)}^{\eta(x,t)} \frac{\partial k}{\partial t} \, \delta\rho \, dz =$$

$$= \frac{\partial}{\partial t} \cancel{\int_{-h(x)}^{\eta(x,t)} k \, \delta\rho \, dz} - [k \, \delta\rho]_{z=\eta} \frac{\partial\eta}{\partial t} - [k \, \delta\rho]_{z=-h} \frac{\partial h}{\partial t} - \int_{-h(x,t)}^{\eta(x,t)} \frac{\partial k}{\partial t} \, \delta\rho \, dz,$$

and:

$$
\begin{aligned}
J_2 \quad = \quad & \int\limits_{-h\,(\boldsymbol{x},t)}^{\eta\,(\boldsymbol{x},t)} \nabla \cdot (k\,\boldsymbol{u}\,\delta\,\rho)\,d\,z \quad - \int\limits_{-h\,(\boldsymbol{x},t)}^{\eta\,(\boldsymbol{x},t)} \nabla \cdot (k\,\boldsymbol{u})\,\delta\,\rho\,d\,z \quad = \\
= \quad & \sum_{i\,=\,1} \int\limits_{-h\,(\boldsymbol{x},t)}^{\eta\,(\boldsymbol{x},t)} \frac{\partial}{\partial\,x_i}\,(k\,u_i\,\delta\,\rho)\,d\,z \;+ \int\limits_{-h\,(\boldsymbol{x},t)}^{\eta\,(\boldsymbol{x},t)} \frac{\partial}{\partial\,z}\,(k\,u_3\,\delta\,\rho)\,d\,z \\
- \quad & \int\limits_{-h\,(\boldsymbol{x},t)}^{\eta\,(\boldsymbol{x},t)} \nabla \cdot (k\,\boldsymbol{u})\,\delta\,\rho\,d\,z \quad = \\
= \quad & \sum_{i\,=\,1}^{2} \left( \frac{\partial}{\partial\,x_i} \int\limits_{-h\,(\boldsymbol{x},t)}^{\eta\,(\boldsymbol{x},t)} k\,u_i\,\delta\,\rho\,d\,z \;-\; [k\,u_i\,\delta\,\rho]_{z\,=\,\eta}\,\frac{\partial\,\eta}{\partial\,x_i} \;-\; [k\,u_i\,\delta\,\rho]_{z\,=\,-h}\,\frac{\partial\,h}{\partial\,x_i} \right) \\
+ \;[k\,u_3\,\delta\,\rho]_{z\,=\,\eta} \;-\; & [k\,u_3\,\delta\,\rho]_{z\,=\,-h} \;-\; \int\limits_{-h\,(\boldsymbol{x},t)}^{\eta\,(\boldsymbol{x},t)} \nabla \cdot (k\,\boldsymbol{u})\,\delta\,\rho\,d\,z.
\end{aligned}
$$

The underlined terms of the last expression, though, are similar to the terms of $I_{\partial V_{\text{lat}}}$, Equation (A4), studied above, and are treated in the same way. Thus, performing this analysis and substituting the results for the integrals $J_1$, $J_2$ into Equation (A6), results, after some simple algebraic manipulations, in Equation (26d).

*Variation with respect to the velocity field:* Regarding the calculation of $\delta_{\boldsymbol{u}}\,(\cdot)$, we start with the variations $\delta_{u_i}\,(\cdot)$, $i\,=\,1,\,2$ of the two horizontal velocity components (repeated indices are not summed/we do not use the summation convention here):

$$
\begin{aligned}
\delta_{u_i}\,\widetilde{\mathscr{S}} \quad = \quad & \int\limits_{T}\int\limits_{D} \int\limits_{-h\,(\boldsymbol{x},t)}^{\eta\,(\boldsymbol{x},t)} \left\{ \rho\,u_i \;-\; k\,\frac{\partial\,\rho}{\partial\,x_i} \;-\; \rho\,\mathbf{A}\,\frac{\partial\,\mathbf{a}}{\partial\,x_i} \right\} \delta\,u_i\,d\,z\,d\,\boldsymbol{x}\,d\,t \\
- \quad & \int\limits_{T}\int\limits_{D} \int\limits_{-h\,(\boldsymbol{x},t)}^{\eta\,(\boldsymbol{x},t)} \rho\,k\,\frac{\partial\,(\delta\,u_i)}{\partial\,x_i}\,d\,z\,d\,\boldsymbol{x}\,d\,t \quad = \\
= \quad & \int\limits_{T}\int\limits_{D} \int\limits_{-h\,(\boldsymbol{x})}^{\eta\,(\boldsymbol{x},t)} \left\{ \rho\,u_i \;-\; k\,\frac{\partial\,\rho}{\partial\,x_i} \;-\; \rho\,\mathbf{A}\,\frac{\partial\,\mathbf{a}}{\partial\,x_i} \;+\; \frac{\partial\,(\rho\,k)}{\partial\,x_i} \right\} \delta\,u_i\,d\,z\,d\,\boldsymbol{x}\,d\,t \\
- \quad & \int\limits_{T}\int\limits_{D} \int\limits_{-h\,(\boldsymbol{x})}^{\eta\,(\boldsymbol{x},t)} \frac{\partial}{\partial\,x_i}\,(\rho\,k\,\delta\,u_i)\,d\,z\,d\,\boldsymbol{x}\,d\,t
\end{aligned}
$$

Hence, exploiting the Leibnitz integral rule, for the treatment of the vertical integral in the last term, the above becomes:

$$
\begin{aligned}
\delta_{u_i}\,\widetilde{\mathscr{S}} \quad = \quad & \int\limits_{T}\int\limits_{D} \Bigg\{ \int\limits_{-h\,(\boldsymbol{x},t)}^{\eta\,(\boldsymbol{x},t)} \left( \rho\,u_i \;-\; k\,\frac{\partial\,\rho}{\partial\,x_i} \;-\; \rho\,\mathbf{A}\,\frac{\partial\,\mathbf{a}}{\partial\,x_i} \;+\; \frac{\partial\,(\rho\,k)}{\partial\,x_i} \right) \delta\,u_i\,d\,z \\
& + [\rho\,k\,\delta\,u_i]_{z\,=\,\eta}\,\frac{\partial\,\eta}{\partial\,x_i} \;+\; [\rho\,k\,\delta\,u_i]_{z\,=\,-h}\,\frac{\partial\,h}{\partial\,x_i} \Bigg\}\,d\,\boldsymbol{x}\,d\,t \\
& \underbrace{ -\; \int\limits_{T}\int\limits_{D} \frac{\partial}{\partial\,x_i} \int\limits_{-h\,(\boldsymbol{x},t)}^{\eta\,(\boldsymbol{x},t)} \rho\,k\,\delta\,u_i\,d\,z\,d\,\boldsymbol{x}\,d\,t }_{K_i}
\end{aligned}
\tag{A7}
$$

Similarly, the calculation of $\delta_{u_3}\,\widetilde{\mathscr{S}}$ yields:

$$\delta_{u_3} \mathscr{F} = \int_T \int_D \int_{-h(x,t)}^{\eta(x,t)} \left\{ \rho \, u_3 - k \frac{\partial \rho}{\partial z} - \rho \, A \frac{\partial a}{\partial z} \right\} \delta \, u_3 \, dz \, dx \, dt$$

$$- \int_T \int_D \int_{-h(x,t)}^{\eta(x,t)} \rho \, k \frac{\partial (\delta \, u_3)}{\partial z} \, dz \, dx \, dt \ =$$

$$= \int_T \int_D \int_{-h(x,t)}^{\eta(x,t)} \left\{ \rho \, u_3 - k \frac{\partial \rho}{\partial z} - \rho \, A \frac{\partial a}{\partial z} + \frac{\partial}{\partial z}(\rho \, k) \right\} \delta \, u_3 \, dz \, dx \, dt$$

$$- \int_T \int_D \int_{-h(x,t)}^{\eta(x,t)} \frac{\partial}{\partial z}(\rho \, k \, \delta \, u_3) \, dz \, dx \, dt \ =$$

$$= \int_T \int_D \int_{-h(x,t)}^{\eta(x,t)} \left\{ \rho \, u_3 - k \frac{\partial \rho}{\partial z} - \rho \, A \frac{\partial a}{\partial z} + \frac{\partial}{\partial z}(\rho \, k) \right\} \delta \, u_3 \, dz \, dx \, dt$$

$$- \int_T \int_D \left( \left[ \rho \, k \, \delta \, u_3 \right]_{z = \eta} - \left[ \rho \, k \, \delta \, u_3 \right]_{z = -h} \right) dx \, dt \,. \tag{A8}$$

Thus, combining Equations (A7) and (A8), and treating the sum of the integrals $K_1$, $K_2$ (last integral of Equation (A7), for $i = 1, 2$) in the same manner as the integral $I_{\partial V_{\text{lat}}}$, Equation (A4), we conclude with Equation (26e).

**Appendix B. Proofs of Lemmata 1 and 2**

In this Appendix, we provide the proofs of the two lemmata that are used in Section 5.
***Proof of Lemma* 1:** Let $\widetilde{f} \equiv [f]_{z=\eta} = f(x, \eta(x, t), t)$. Then, using the chain rule:

$$\nabla_2 \widetilde{f} = \left( \frac{\partial_e \widetilde{f}}{\partial x_1} + \frac{\partial \widetilde{f}}{\partial \eta} \frac{\partial \eta}{\partial x_1}, \frac{\partial_e \widetilde{f}}{\partial x_2} + \frac{\partial \widetilde{f}}{\partial \eta} \frac{\partial \eta}{\partial x_2} \right), \tag{A9}$$

where $\partial_e (\cdot) / \partial x_i$ differs from $\partial (\cdot) / \partial x_i$ in the sense that it acts only on the explicit dependence of $\widetilde{f}$ on $x_i$; $\widetilde{f}$ depends on $t$, $x_1$, and $x_2$ both explicitly and implicitly via $\eta(x, t)$. Additionally, it may easily be checked that:

$$\frac{\partial_e \widetilde{f}}{\partial x_i} = \left[ \frac{\partial f(x, z, t)}{\partial x_i} \right]_{z=\eta}, \ i = 1, 2, \text{ and } \frac{\partial \widetilde{f}}{\partial \eta} = \left[ \frac{\partial f(x, z, t)}{\partial z} \right]_{z=\eta}. \tag{A10}$$

Thus, substituting Equations (A10) into Equation (A9), we find:

$$\nabla_2 \widetilde{f} = \left( \left[ \frac{\partial f}{\partial x_1} \right]_{z=\eta} + \left[ \frac{\partial f}{\partial z} \right]_{z=\eta} \frac{\partial \eta}{\partial x_1}, \left[ \frac{\partial f}{\partial x_2} \right]_{z=\eta} + \left[ \frac{\partial f}{\partial z} \right]_{z=\eta} \frac{\partial \eta}{\partial x_2} \right). \tag{A11}$$

Combining the left-hand side of Lemma 1, Section 5.3, with the above relation, we obtain:

$$\nabla_2 \left( [f]_{z=\eta} \right) (\delta B_{\eta, 1}, \delta B_{\eta, 2}) =$$
$$= \left( \left[ \frac{\partial f}{\partial x_1} \right]_{z=\eta} + \left[ \frac{\partial f}{\partial z} \right]_{z=\eta} \frac{\partial \eta}{\partial x_1} \right) \delta B_{\eta, 1} + \left( \left[ \frac{\partial f}{\partial x_2} \right]_{z=\eta} + \left[ \frac{\partial f}{\partial z} \right]_{z=\eta} \frac{\partial \eta}{\partial x_2} \right) \delta B_{\eta, 2} =$$
$$= \left[ \frac{\partial f}{\partial x_1} \right]_{z=\eta} \delta B_{\eta, 1} + \left[ \frac{\partial f}{\partial x_2} \right]_{z=\eta} \delta B_{\eta, 2} + \left[ \frac{\partial f}{\partial z} \right]_{z=\eta} \left( \frac{\partial \eta}{\partial x_1} \delta B_{\eta, 1} + \frac{\partial \eta}{\partial x_2} \delta B_{\eta, 2} \right) = \tag{A12}$$
$$= [\nabla f]_{z=\eta} \cdot \left( \delta B_{\eta, 1}, \delta B_{\eta, 2}, \frac{\partial \eta}{\partial x_1} \delta B_{\eta, 1} + \frac{\partial \eta}{\partial x_2} \delta B_{\eta, 2} \right).$$

Additionally, recalling Equations (45a) and (45b):

$$\left( \delta B_{\eta, 1}, \ \delta B_{\eta, 2}, \ \frac{\partial \eta}{\partial x_1} \delta B_{\eta, 1} + \frac{\partial \eta}{\partial x_2} \delta B_{\eta, 2} \right) = \delta B_{\eta, 1} T_{\eta, 1} + \delta B_{\eta, 2} T_{\eta, 2}. \tag{A13}$$

Using Equation (A13) in Equation (A12) concludes the proof of the lemma.

***Proof of Lemma 2:*** Let $\tilde{f} \equiv [f]_{z=\eta} = f(\boldsymbol{x}, \eta(\boldsymbol{x}, t), t)$. Then, the above Equations (A9)–(A11) hold, along with:

$$
\begin{aligned}
\frac{\partial \tilde{f}}{\partial t} &= \frac{\partial}{\partial t}\left([f]_{z=\eta}\right) = \frac{\partial_e \tilde{f}}{\partial t} + \frac{\partial \tilde{f}}{\partial \eta}\frac{\partial \eta}{\partial t} = \\
&= \left[\frac{\partial f(\boldsymbol{x}, z, t)}{\partial t}\right]_{z=\eta} + \left[\frac{\partial f(\boldsymbol{x}, z, t)}{\partial z}\right]_{z=\eta}\frac{\partial \eta}{\partial t},
\end{aligned}
\tag{A14}
$$

where $\partial_e(\cdot)/\partial t$ has a similar meaning to $\partial_e(\cdot)/\partial x_i$ in the proof of Lemma 1. Further, recalling that $\boldsymbol{u}_{2\,D} = (u_1, u_2)$:

$$
\nabla_2 \cdot \left([f\,\boldsymbol{u}_{2\,D}]_{z=\eta}\right) = \frac{\partial}{\partial x_1}\left([f\,u_1]_{z=\eta}\right) + \frac{\partial}{\partial x_2}\left([f\,u_2]_{z=\eta}\right),
$$

which, based on Equations (A9) and (A10), is written as:

$$
\begin{aligned}
\nabla_2 \cdot &\left([f\,\boldsymbol{u}_{2D}]_{z=\eta}\right) = \\
&= \left[\frac{\partial(f\,u_1)}{\partial x_1}\right]_{z=\eta} + \left[\frac{\partial(f\,u_1)}{\partial z}\right]_{z=\eta}\frac{\partial \eta}{\partial x_1} + \left[\frac{\partial(f\,u_2)}{\partial x_2}\right]_{z=\eta} + \left[\frac{\partial(f\,u_2)}{\partial z}\right]_{z=\eta}\frac{\partial \eta}{\partial x_2} = \\
&= \left[\frac{\partial(f\,u_1)}{\partial x_1} + \frac{\partial(f\,u_2)}{\partial x_2}\right]_{z=\eta} + \left[\frac{\partial f}{\partial z}\right]_{z=\eta}\left([u_1]_{z=\eta}\frac{\partial \eta}{\partial x_1} + [u_2]_{z=\eta}\frac{\partial \eta}{\partial x_2}\right) \\
&+ [f]_{z=\eta}\left(\left[\frac{\partial u_1}{\partial z}\right]_{z=\eta}\frac{\partial \eta}{\partial x_1} + \left[\frac{\partial u_2}{\partial z}\right]_{z=\eta}\frac{\partial \eta}{\partial x_2}\right)
\end{aligned}
\tag{A15}
$$

Accordingly, combining Equations (A14) and (A15), the left-hand side of Lemma 2, Section 5.3, is equal to:

$$
\begin{aligned}
\frac{\partial}{\partial t}\left([f]_{z=\eta}\right) + \nabla_2 \cdot &\left([f\,\boldsymbol{u}_{2D}]_{z=\eta}\right) = \\
&= \left[\frac{\partial f}{\partial t}\right]_{z=\eta} + \left[\frac{\partial(f\,u_1)}{\partial x_1} + \frac{\partial(f\,u_2)}{\partial x_2}\right]_{z=\eta} \\
&+ \left[\frac{\partial f}{\partial z}\right]_{z=\eta}\underbrace{\left(\frac{\partial \eta}{\partial t} + [u_1]_{z=\eta}\frac{\partial \eta}{\partial x_1} + [u_2]_{z=\eta}\frac{\partial \eta}{\partial x_2}\right)}_{Q_1} \\
&+ [f]_{z=\eta}\underbrace{\left(\left[\frac{\partial u_1}{\partial z}\right]_{z=\eta}\frac{\partial \eta}{\partial x_1} + \left[\frac{\partial u_2}{\partial z}\right]_{z=\eta}\frac{\partial \eta}{\partial x_2}\right)}_{Q_2}
\end{aligned}
\tag{A16}
$$

Now, invoking the free-surface kinematic condition, Equation (49), we find that:

$$
Q_1 = [u_3]_{z=\eta} \text{ and } Q_2 = \left[\frac{\partial u_3}{\partial z}\right]_{z=\eta},
$$

where the second relation is derived by differentiating the kinematic condition with respect to $z$. Using those relations in Equation (A16) and rearranging the terms, we obtain:

$$
\frac{\partial}{\partial t}\left([f]_{z=\eta}\right) + \nabla_2 \cdot \left([f\,\boldsymbol{u}_{2\,D}]_{z=\eta}\right) = \left[\frac{\partial f}{\partial t} + \nabla \cdot (f\,\boldsymbol{u})\right]_{z=\eta},
$$

completing the proof.

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
