# Peer review of "Hamiltonian Variational Formulation of Three-Dimensional, Rotational Free-Surface Flows, with a Moving Seabed, in the Eulerian Description"

_fluids, doi:10.3390/fluids7100327_

Round 1

Reviewer 1 Report

The novelty of the present work is to be clearly stated.

What are the advantages of the presented formulation compared with the 3D vector potential-vorticity formulation that can deal easily with such problesm?

Eq. 12 is to be checked.

 It is not sufficient to discuss the problem only from mathematical point of view.

Some solutions of the proposed problem for specific cases are to be presented to check the validity of the formulation.

The equations are to be checked, there some misprints.

The English level is to be improved.

Author Response

Please, find our reply in the attached file.

Reviewer 2 Report

In this study, the authors provided a complete derivation of the equations of motion and of the boundary conditions for 3D rotational flows with a free surface and a moving seabed, by means of Hamilton’s Principle. They considered that variational procedure succeeds in two main points: Firstly, besides the derivation of the full equations for the bulk fluid motion, convenient representations of the velocity and pressure fields are obtained in terms of Clebsch-Lin potentials; Secondly, re-expressing the boundary variational equation in terms of the boundary fluid parcels’ virtual displacements, leads to a successful derivation of both kinematic and dynamic boundary conditions. Although the authors adopted many pages to derive these equations, the verification results with other theoretical or laboratory data are not found. This paper is only shown the derivation of equations of motion and of the boundary conditions. Moreover, the authors revealed that ” two certain concepts are involved require further clarification: i) the two level constraints (different in the fluid volume and on the fluid boundary), introduced herein for the first time, call for a better understanding and theoretical assessment; ii) the gauge freedom of the Clebsch-Lin potentials and the ability of the latter to represent generic rotational flows need to be clarified. It is the authors’ intention to address those issues in the near future.” It is difficult to find the highlights of this paper. Hence, this manuscript is incomplete and the theoretical derivations need to resolve the problems. I cannot see how this article meets the standards of the Fluids, therefore, I recommend rejection.

Author Response

(The authors gave the same response as above.)

Reviewer 3 Report

An excellent paper. 

Minor corrections:

page 2: non convenient --> incovenient

page 38: Proof of Lemma 2: parentheses are missing

Author Response

We would like to thank this reviewer for his/her appreciation and positive feedback concerning our work. The two suggested corrections have been implemented in the revised version.

Round 2

Reviewer 1 Report

After revision, the paper can be accepted for publication

Author Response

We would like to thank this reviewer for his/her time and effort to review our paper.

Reviewer 2 Report

The revised version of the paper has been modified based on the reviewer' comments and its content is clear to show the “Purpose” “Highlight” and “Contribution” in the research condition. It has been summarized systematically, however, this paper uses too much literature. Generally, the manuscript met the publication criteria and objective of Fluids; acceptance is suggested after the authors remove some very old or unnecessary reference.

Author Response

We would like to thank this reviewer for his/her time and effort to review our paper. A reasonable reduction of the no. of references has been made, as you suggested.